# Gradient-based Bi-level Optimization for Deep Learning: A Survey

## Abstract

Bi-level optimization, especially the gradient-based category, has been widely used in the deep learning community including hyperparameter optimization and meta-knowledge extraction. Bi-level optimization embeds one problem within another and the gradient-based category solves the outer-level task by computing the hypergradient, which is much more efficient than classical methods such as the evolutionary algorithm. In this survey, we first give a formal definition of the gradient-based bi-level optimization. Next, we delineate criteria to determine if a research problem is apt for bi-level optimization and provide a practical guide on structuring such problems into a bi-level optimization framework, a feature particularly beneficial for those new to this domain. More specifically, there are two formulations: the single-task formulation to optimize hyperparameters such as regularization parameters and the distilled data, and the multi-task formulation to extract meta-knowledge such as the model initialization. With a bi-level formulation, we then discuss four bi-level optimization solvers to update the outer variable including explicit gradient update, proxy update, implicit function update, and closed-form update. Finally, we wrap up the survey by highlighting two prospective future directions: (1) *Effecctive Data Optimization for Science* examined through the lens of task formulation. (2) *Accurate Explicit Proxy Update* analyzed from an optimization standpoint.

## 1 Introduction

With the fast development of deep learning, bi-level optimization is drawing lots of research attention due to the nested problem structure in many deep learning problems, including hyperparameter optimization (Rendle, 2012; Chen et al., 2019; Liu et al., 2019) and meta-knowledge extraction (Finn et al., 2017). The bi-level optimization problem is a special kind of optimization problem where one problem is embedded within another and can be traced to two domains: one is from game theory where the leader and the follower compete on quantity in the Stackelberg game (Von Stackelberg, 2010); another one is from mathematical programming where the inner level problem serves as a constraint on the outer level problem (Bracken & McGill, 1973). Especially, compared with classical methods (Sinha et al., 2017) which require strict mathematical properties or can not scale to large datasets, the efficient gradient descent methods provide a promising solution to the complicated bi-level optimization problem and thus are widely adopted in much deep learning research work to optimize hyperparameters in the single-task formulation (Bertinetto et al., 2019; Hu et al., 2019; Liu et al., 2019; Rendle, 2012; Chen et al., 2019; Ma et al., 2020; Zhang et al., 2023; Li et al., 2022) or extract meta-knowledge in the multi-task formulation (Finn et al., 2017; Andrychowicz et al., 2016; Chen et al., 2023b; Zhong et al., 2022; Chi et al., 2021; 2022; Wu et al., 2022; Chen et al., 2022c).

In this survey, we mainly focus on gradient-based bi-level optimization regarding deep neural networks with an explicitly defined objective function. This survey aims to guide researchers on their research problems involving bi-level optimization. We first define notations and give a formal definition of gradient-based bi-level optimization in Section 2. We then propose a new taxonomy in terms of task formulation in Section 3 and methods to compute the hypergradient of the outer variable in Section 4. This taxonomy provides guidance to researchers on the criteria and procedures to formulate a task as a bi-level optimization problem and how to solve this problem. Last, we conclude the survey with two promising future directions in Section 5.

## 2 Definition

Table 1: Key notations used in this paper.

| Notations | Descriptions |
|---|---|
| $\boldsymbol{x}_i$ | Input of data point indexed by i |
| $\boldsymbol{y}_i$ | Label of data point indexed by i |
| $\mathcal{D}/\mathcal{D}_{train}/\mathcal{D}_{val}$ | Supervised/Training/Validation dataset |
| $|\mathcal{D}|$ | Number of samples in the dataset $\mathcal{D}$ |
| $\boldsymbol{\theta}/\boldsymbol{\Theta}$ | Inner learnable variable |
| $\boldsymbol{\phi}/\boldsymbol{\Phi}$ | Outer learnable variable |
| $\boldsymbol{\theta}^*(\boldsymbol{\phi})$ | Best response of $\boldsymbol{\theta}$ given $\boldsymbol{\phi}$ |
| $l(\boldsymbol{\theta}, \boldsymbol{\phi}, \boldsymbol{x}_i, y_i)$ | Loss on the $i_{th}$ data point $\boldsymbol{x}_i, \boldsymbol{y}_i$ |
| $l(\boldsymbol{\theta}, \boldsymbol{x}_i, y_i)$ | Loss on data $\boldsymbol{x}_i, \boldsymbol{y}_i$ without $\boldsymbol{\phi}$ |
| $\mathcal{L}(\boldsymbol{\theta}, \boldsymbol{\phi}, \mathcal{D})$ | Loss on the whole dataset $\mathcal{D}$ |
| $\mathcal{L}(\boldsymbol{\theta}, \mathcal{D})$ | Loss on dataset $\mathcal{D}$ without $\boldsymbol{\phi}$ |
| $\mathcal{L}^{in}(\boldsymbol{\theta}, \boldsymbol{\phi}, \mathcal{D})$ | Inner level loss on dataset $\mathcal{D}$ |
| $\mathcal{L}^{out}(\boldsymbol{\theta}, \boldsymbol{\phi}, \mathcal{D})$ | Outer level loss on dataset $\mathcal{D}$ |
| $\frac{d\mathcal{L}^{out}}{d\boldsymbol{\phi}}$ | Hypergradient regarding $\boldsymbol{\phi}$ |
| $\eta$ | Inner level learning rate |
| M | Number of inner level tasks |
| $\Omega(\boldsymbol{\theta}, \boldsymbol{\phi})$ | Regularization parameterized by $\boldsymbol{\phi}$ |
| OPT | Some optimizer like Adam |
| $\mathcal{D}_{real}/\mathcal{D}_{syn}$ | Real/Synthetic dataset |
| $p(\cdot)$ | Product price in the market |
| $C_l(\phi)/C_f(\theta)$ | Cost of leader and follower |
| $D$ | Predicted atom-atom distances |
| $\mathcal{D}^{prf}/\mathcal{D}^{ft}$ | Pretraining/finetuning data |
| $E$ | Some equation constraint |
| $\lambda/\gamma$ | Regularization strength parameter |
| $\epsilon/\tau$ | Some small positive constant |
| $P_{\boldsymbol{\alpha}}$ | Proxy network parameterized by $\boldsymbol{\alpha}$ |
| $T$ | Number of iterations in optimization |

In this section, we define the gradient-based bi-level optimization, which focuses on neural networks with an explicit objective function. For convenience, we list notations and their descriptions in Table 1.

Assume there is a dataset $\mathcal{D} = \{(\boldsymbol{x}_i, y_i)\}$ under the supervised learning setting where $\boldsymbol{x}_i$ and $y_i$ represent the $i^{th}$ input and corresponding label, respectively. Besides, $\mathcal{D}_{train}$ and $\mathcal{D}_{val}$ represent the training set and the validation set, respectively. We use $\boldsymbol{\theta} \in \mathbb{R}^d$ ($\boldsymbol{\Theta}$ for matrix form) to parameterize the inner learnable variable which often refers to the model parameters and use $\boldsymbol{\phi} \in \mathbb{R}^m$ ($\boldsymbol{\Phi}$ for matrix form) to parameterize the outer learnable variable including the hyperparameters and the meta knowledge. In this paper, the hyperparameters are not limited to the regularization and the learning rate but refer to any knowledge in a single task formulation, as we will illustrate more detailedly in Section 3. Denote the loss function on $(\boldsymbol{x}_i, y_i)$ as $l(\boldsymbol{\theta}, \boldsymbol{x}_i, y_i)$, which refers to a certain format of objectives depending on the tasks such as Cross-Entropy loss or Mean Square Error (MSE) loss. Note that in some cases we use $l(\boldsymbol{\theta}, \boldsymbol{\phi}, \boldsymbol{x}_i, y_i)$, which is an equivalent variant of $l(\boldsymbol{\theta}, \boldsymbol{x}_i, y_i)$ under this setting. This is because the outer learnable parameters $\boldsymbol{\phi}$ can be hyperparameters like the learning rate when calculating $l(\boldsymbol{\theta}, \boldsymbol{x}_i, y_i)$ and is thus not explicitly represented. We then use $\mathcal{L}(\boldsymbol{\theta}, \boldsymbol{\phi}, \mathcal{D})$ or $\mathcal{L}(\boldsymbol{\theta}, \mathcal{D})$ to denote the loss over the dataset $\mathcal{D}$, and represent the inner level loss and the outer level loss as $\mathcal{L}^{in}(\boldsymbol{\theta}, \boldsymbol{\phi}, \mathcal{D})$ and $\mathcal{L}^{out}(\boldsymbol{\theta}, \boldsymbol{\phi}, \mathcal{D})$, respectively. Generally, the inner level loss is computed on the training dataset $\mathcal{D}^{train}$, and the outer level loss is assessed on the validation dataset $\mathcal{D}^{val}$. We use $\eta$ to represent the learning rate adopted by the inner-level optimization. Employing these notations, we present the mathematical expression for the bi-level optimization problem as follows:

$$\phi^* = \arg\min_{\phi} \mathcal{L}^{out}(\theta^*(\phi), \phi). \tag{1}$$

$$\text{s.t.} \quad \theta^*(\phi) = \arg\min_{\theta} \mathcal{L}^{in}(\theta, \phi). \tag{2}$$

The inner level problem in Eq. (2) serves as a constraint and builds the relation between the $\phi$ and $\theta$. Here we use the arg min form in Eq. (2) but note that the inner level task can be extended to some equation constraints as we will further illustrate in Section 3. In the nature of neural networks, one can use gradient descent to estimate $\theta^*(\phi)$. The outer level problem acts as the main optimization problem and computes the hypergradient $\frac{d\mathcal{L}^{out}}{d\phi}$ to update the outer variable $\phi$ by leveraging the relation built in Eq. (2),

$$\frac{d\mathcal{L}^{out}}{d\phi} = \frac{\partial \mathcal{L}^{out}}{\partial \theta} \frac{\partial \theta(\phi)}{\partial \phi} + \frac{\partial \mathcal{L}^{out}}{\partial \phi}. \tag{3}$$

This is called *gradient-based bi-level optimization*. When extended to the multi-task scenario to extract meta-knowledge on $M$ different tasks, the above formulation can be rewritten as:

$$\phi^* = \arg\min_{\phi} \sum_{i=1}^{M} \mathcal{L}^{out}(\theta_i^*(\phi), \phi). \tag{4}$$

$$\text{s.t.} \quad \theta_i^*(\phi) = \arg\min_{\theta} \mathcal{L}_i^{in}(\theta, \phi). \tag{5}$$

Here $\phi$ symbolizes the meta-knowledge, denoting the knowledge that spans across multiple tasks, exemplified by aspects such as the model initialization.

## 3 Task Formulation

The classification of bi-level optimization task formulation into single-task and multi-task types, as portrayed in Figure 1, hinges on the kind of knowledge we are aiming to learn (Franceschi et al., 2018). The single-task formulation focuses on the learning of hyperparameters within a single task, while the multi-task formulation is oriented towards the acquisition of meta-knowledge. These two formulations are expounded upon in Sections 3.1 and 3.2, respectively.

### 3.1 Single-task formulation

The single-task formulation applies bi-level optimization on a single task and aims to learn hyperparameters for the task. Note that in this paper, the meaning of hyperparameter is not limited to its traditional meaning like regularization but has a broader meaning, referring to all single-task knowledge.

A particular single-task problem can be deemed suitable for bi-level optimization if it meets two criteria. Firstly, it has a main optimization problem guiding the optimization of the outer variable. Secondly, a constraint exists between the inner and outer variables such that a differentiable relationship between these variables can be established. To elaborate, our first step is to identify the inner variable, denoted as $\theta$, and the outer variable, $\phi$. Next, we identify the main optimization component which optimizes the hyperparameters, which acts as the outer level problem. Finally, the inner level problem is framed by recognizing the constraint between these two variables, which further enables us to establish a differentiable relationship between them.

(1) A notable situation is when the constraint and the main optimization problem use different mathematical formulas, implying that they optimize entirely different problems. In these scenarios, the constraint typically arises organically, and its identification becomes straightforward. Examples include the energy constraint present in topology design (Christiansen et al., 2001) or the bio-chemical constraint in protein representation learning (Chen et al., 2022b). Conversely, when the main optimization problem and the constraint share the same mathematical formula, the formula needs to be broken down into two levels. This breakdown is usually accomplished by considering data variations between training and validation sets. The inner level,

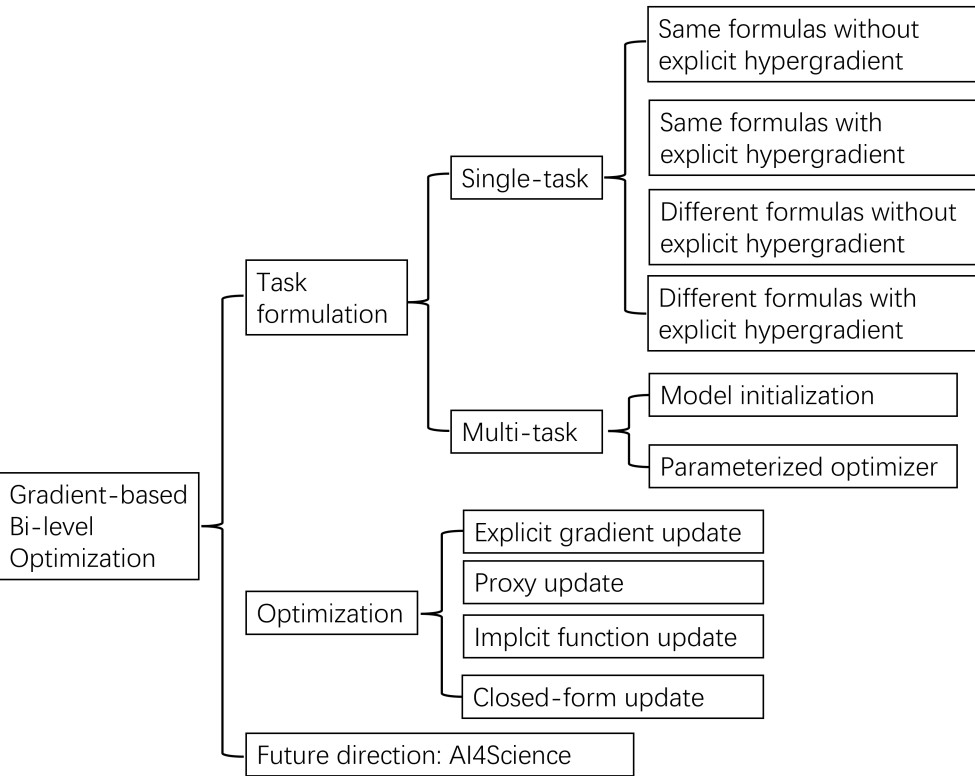

Figure 1: Summary of gradient-based bi-level optimization.

typically represented by the training loss, functions as the constraint in this setting (Franceschi et al., 2018). This comprises the first criterion for our evaluation. (2) In some cases, the main optimization problem might not directly contain the outer variable. In such cases, the connection built at the inner level is utilized. This situation poses a challenge in formulating the outer level task, thereby leading us to introduce a second criterion. This second criterion classifies works based on whether the calculation of the hypergradient relies exclusively on the established inner level connection, $\frac{d\mathcal{L}^{out}(\boldsymbol{\theta}^*(\boldsymbol{\phi}))}{d\boldsymbol{\phi}}$, or not $\frac{d\mathcal{L}^{out}(\boldsymbol{\theta}^*(\boldsymbol{\phi}),\boldsymbol{\phi})}{d\boldsymbol{\phi}}$.

To sum up, we consider the following four cases and discuss the corresponding examples for better illustration:

- 1). $\mathcal{L}^{in}$ and $\mathcal{L}^{out}$ share the same mathematical formula and the hypergradient only comes from the inner level connection $\boldsymbol{\theta}(\boldsymbol{\phi})$;

- 2). $\mathcal{L}^{in}$ and $\mathcal{L}^{out}$ share the same mathematical formula and the hypergradient comes from both the inner level connection $\boldsymbol{\theta}(\boldsymbol{\phi})$ and the outer level objective explicitly;

- 3). $\mathcal{L}^{in}$ and $\mathcal{L}^{out}$ have different mathematical formulas and the hypergradient only comes from the inner level connection $\boldsymbol{\theta}(\boldsymbol{\phi})$;

- 4). $\mathcal{L}^{in}$ and $\mathcal{L}^{out}$ have different mathematical formulas and the hypergradient comes from both the inner level connection $\boldsymbol{\theta}(\boldsymbol{\phi})$ and the outer level objective explicitly;

### 3.1.1 Same formula without explicit outer level hypergradient

In instances where the inner and outer levels utilize the same mathematical formula, the objective of the task is the optimization of a single goal, viewed from two distinct perspectives. The outer variable often describes some aspects of the training process besides model parameters. The outer variable does not manifest explicitly in the main optimization problem, leading to a lack of explicit hypergradient at the outer

Table 2: Same formula without explicit outer level hypergradient.

| Work | Inner Var $\boldsymbol{\theta}$ | Outer Var $\boldsymbol{\phi}$ | Inner Level Problem as Constraint | Outer Level Problem as Main Opt |
|---|---|---|---|---|
| (a) | Model params | Regularization | $\boldsymbol{\theta}^*(\boldsymbol{\phi}) = \arg\min_{\boldsymbol{\theta}} \sum_{(\boldsymbol{x}_i,y_i)\in\mathcal{D}_{train}} l(\boldsymbol{\theta},\boldsymbol{x}_i,\boldsymbol{y}_i) + \Omega(\boldsymbol{\theta},\boldsymbol{\phi})$ | $\arg\min_{\boldsymbol{\phi}} \sum_{(\boldsymbol{x}_i,y_i)\in\mathcal{D}_{val}} l(\boldsymbol{\theta}^*(\boldsymbol{\phi}),\boldsymbol{x}_i,\boldsymbol{y}_i)$ |
| (b) | Model params | Learning rate | $\boldsymbol{\theta}^*(\boldsymbol{\phi}) = \text{OPT}(\boldsymbol{\theta},\boldsymbol{\phi},\frac{\partial\mathcal{L}(\boldsymbol{\theta},\mathcal{D}_{train})}{\partial\boldsymbol{\theta}})$ | $\arg\min_{\boldsymbol{\phi}} \sum_{(\boldsymbol{x}_i,y_i)\in\mathcal{D}_{val}} l(\boldsymbol{\theta}^*(\boldsymbol{\phi}),\boldsymbol{x}_i,\boldsymbol{y}_i)$ |
| (c) | Model params | Perturbation | $\boldsymbol{\theta}^*(\boldsymbol{\phi}) = \arg\min_{\boldsymbol{\theta}} \sum_{(\boldsymbol{x}_i,y_i)\in\mathcal{D}_{train}} l(\boldsymbol{\theta},\boldsymbol{x}_i+\boldsymbol{\phi}_i,\boldsymbol{y}_i).$ | $\arg\min_{\boldsymbol{\phi}} \sum_{(\boldsymbol{x}_j,\boldsymbol{y}_j)\in\mathcal{D}_{val}} -l(\boldsymbol{\theta}^*(\boldsymbol{\phi}),\boldsymbol{x}_j|\boldsymbol{y}_j)$ |
| (d) | Model params | Distilled data | $\boldsymbol{\theta}^*(\boldsymbol{\phi}) = \arg\min_{\boldsymbol{\theta}} \sum_{(\boldsymbol{\phi}_i,\boldsymbol{y}_i)\in\mathcal{D}_{syn}} l(\boldsymbol{\theta},\boldsymbol{\phi}_i,\boldsymbol{y}_i)$ | $\arg\min_{\boldsymbol{\phi}} \sum_{(\boldsymbol{x}_j,\boldsymbol{y}_j)\in\mathcal{D}_{real}} l(\boldsymbol{\theta}^*(\boldsymbol{\phi}),\boldsymbol{x}_j,\boldsymbol{y}_j)$ |
| (e) | Model params | Data label | $\boldsymbol{\theta}^*(\boldsymbol{\phi}) = \arg\min_{\boldsymbol{\theta}} \sum_{(\boldsymbol{x}_i,\boldsymbol{\phi}_i)\in\mathcal{D}_{train}} l(\boldsymbol{\theta},\boldsymbol{x}_i,\boldsymbol{\phi}_i)$ | $\arg\min_{\boldsymbol{\phi}} \sum_{(\boldsymbol{x}_j,\boldsymbol{y}_j)\in\mathcal{D}_{val}} l(\boldsymbol{\theta}^*(\boldsymbol{\phi}),\boldsymbol{x}_j,\boldsymbol{y}_j)$ |
| (f) | Model params | Sample weight | $\boldsymbol{\theta}^*(\boldsymbol{\phi}) = \arg\min_{\boldsymbol{\theta}} \sum_{(\boldsymbol{x}_i,\boldsymbol{y}_i)\in\mathcal{D}_{train}} \phi_i l(\boldsymbol{\theta},\boldsymbol{x}_i,\boldsymbol{y}_i)$ | $\arg\min_{\boldsymbol{\phi}} \sum_{(\boldsymbol{x}_i,\boldsymbol{y}_i)\in\mathcal{D}_{val}} l(\boldsymbol{\theta}^*(\boldsymbol{\phi}),\boldsymbol{x}_i,\boldsymbol{y}_i)$ |

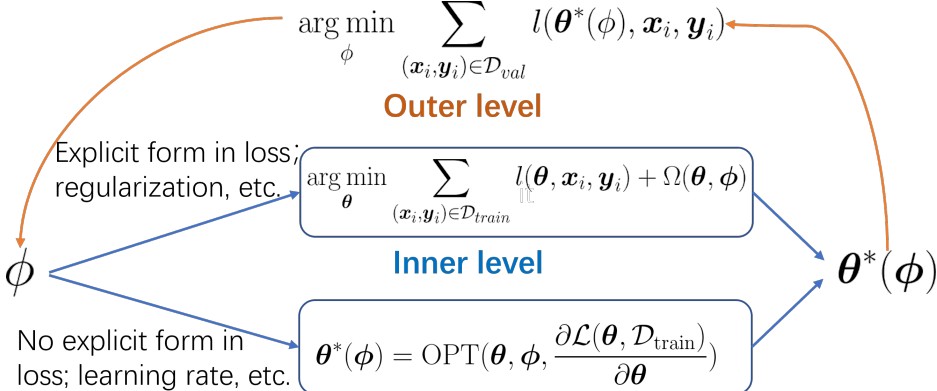

Figure 2: Model-related outer variables.

level. In this case, the hypergradient of $\frac{d\mathcal{L}^{out}}{d\boldsymbol{\phi}}$ is computed through the inner level connection as $\frac{\partial\mathcal{L}^{out}}{\partial\boldsymbol{\theta}}\frac{\partial\boldsymbol{\theta}(\boldsymbol{\phi})^\top}{\partial\boldsymbol{\phi}}$. Depending on the specific implications of $\boldsymbol{\phi}$, outer variables can be divided into two categories: model-related outer variables and data-related outer variables.

**Model-related.** Model-related outer variables often describe the model optimization process, including (a) regularization parameters (Franceschi et al., 2018), (b) learning rate (Franceschi et al., 2017), etc, which are more common compared with data-related ones.

(a) Regularization is an essential component to avoid overfitting in machine learning models. However, identifying an effective regularization term is a challenging task, primarily because each evaluation of a single regularization term necessitates training the entire model. A model trained with a suitable regularization term is expected to produce a low error rate on the validation set. This observation leads Franceschi et al. (2018) to approach the selection of regularization as a bi-level optimization problem to enable direct and efficient optimization. As illustrated in Table 2(a), the inner level loss on the training set acts as a constraint, establishing a differentiable connection between the model parameters $\theta$ and the regularization term. The outer level loss on the validation set forms the main optimization problem, aimed at optimizing the regularization through the inner level connection. In this context, $\Omega(\boldsymbol{\theta},\boldsymbol{\phi})$ denotes the regularization term parameterized by $\boldsymbol{\phi}$ on $\boldsymbol{\theta}$. An example of a simple case is the L2 regularization, where $\Omega(\boldsymbol{\theta},\phi) = \phi|\boldsymbol{\theta}|^2$. In the outer level loss, the regularization term is considered as zero, making it the same as the inner level loss in mathematical form. When dealing with high dimensionalities, traditional methods such as random search and bayesian optimization can prove inadequate. In contrast, the bi-level optimization framework offers an efficient approach to directly update high-dimensional hyperparameters, such as regularization, as demonstrated by Rendle (2012), Chen et al. (2019), and Lorraine et al. (2020).

(b) Contrary to regularization parameters, which are explicitly present in the loss objective, some model-related outer variables exist only within the optimization process. The distinction between these two categories of variables is listed in Figure 2. One example of such a variable is the learning rate, as discussed by Franceschi et al. (2017). The approach to optimize the learning rate, as demonstrated in Table 2(b), resembles the regularization optimization process in the bi-level optimization context. The key difference lies in the way the differentiable connection is constructed. This connection is established by $\boldsymbol{\theta}^*(\boldsymbol{\phi}) = \text{OPT}(\boldsymbol{\theta},\boldsymbol{\phi},\frac{\partial\mathcal{L}(\boldsymbol{\theta},\mathcal{D}_{train})}{\partial\boldsymbol{\theta}})$. Here, OPT represents an optimization process aimed at minimizing the training loss, which acts as the constraint. A simple case can be represented by a one-step Stochastic Gra-

dient Descent (SGD), written as $\boldsymbol{\theta}^*(\boldsymbol{\phi}) = \boldsymbol{\theta} - \phi \frac{\partial \mathcal{L}(\boldsymbol{\theta}, \mathcal{D}_{train})}{\partial \boldsymbol{\theta}^\top}$. In this equation, the outer variable $\phi$ represents the learning rate $\eta$. Though the inner level problem is represented through the perspective of an optimizer, fundamentally it is still the same math objective being optimized as in the outer level loss. This learning rate-learning procedure exhibits some similarities to meta-learned optimization, which we will discuss further in Section 3.2.2.

**Data-related.** Transitioning from the discussion on model-related outer variables, we now turn our attention to data-related variables. As illustrated in Figure 3, either the data point $(\boldsymbol{x}_i, \boldsymbol{y}_i)$ itself or its associated weights can be treated as the outer variable. These can be updated via bi-level optimization across a spectrum of research areas. This includes (c) adversarial attack, (d) data distillation, (e) label learning, and (f) sample reweighting, among others.

(c) Adversarial attacks represent an effort to identify data perturbations, denoted as $\boldsymbol{\phi}$, that lead the model to perform poorly on the validation set. This concept is further elucidated in Table 2(c), showcasing how it forms a bi-level optimization problem (Biggio et al., 2012; Yuan & Wu, 2021). In this context, $\boldsymbol{\phi}_i$ signifies the perturbation added to the sample $\boldsymbol{x}_i$. The inner level acts as a constraint, creating a differentiable connection between these perturbations $\boldsymbol{\phi}_i$ and the model parameters $\boldsymbol{\theta}$. This connection is achieved by minimizing the training loss. Concurrently, the outer level updates the perturbation $\boldsymbol{\phi}$ by maximizing the validation loss. Through this, we can effectively pinpoint adversarial attacks, represented by $\boldsymbol{\phi}_i$. The outer level loss, with zero perturbation, can be viewed as being the same mathmatical form as the inner level loss.

(d) Data distillation techniques (Wang et al., 2018; Lei & Tao, 2023) aim to encapsulate the knowledge from a large training dataset into a significantly smaller one for the purpose of compression. This objective is accomplished by training a model on the smaller dataset and expecting it to deliver strong performance on the larger one, an approach that aligns with the framework of a bi-level optimization problem as demonstrated in Table 2(d). Within this context, $\mathcal{D}_{real}$ is the dataset consisting of real data, whereas $\mathcal{D}_{syn}$ is the dataset made up of synthesized data. The inner level establishes the connection between the $i^{th}$ data point $\boldsymbol{\phi}_i$ and the model parameters $\boldsymbol{\theta}$ by minimizing the training loss. Simultaneously, the outer optimization level updates the synthesized $\boldsymbol{\phi}$ to ensure effective performance on the real data, serving as the main optimization component. Intriguingly, Nguyen et al. (2020) leverage the correspondence between infinitely-wide neural networks and kernels to achieve remarkable results in data distillation. Furthermore, the concept of data distillation has been effectively applied to black-box optimization, as demonstrated by Chen et al. (2022a) and Chen et al. (2023a), yielding impressive outcomes.

(e) Label learning strategies, such as those proposed by Algan & Ulusoy (2021) and Wu et al. (2021), consider the label $\boldsymbol{y_i}$ as an outer variable that is parameterized by $\boldsymbol{\phi}$. Distinct from data distillation methods, label learning strategies do not endeavor to reduce the size of the dataset. The principal aim of this learning approach is to learn cleaner labels that can enhance model performance under label noise, with the guidance of a clean dataset. This process can be conceptualized as a bi-level optimization problem as laid out in Table 2 (e), where the main optimization task is to optimize the label against the validation loss.

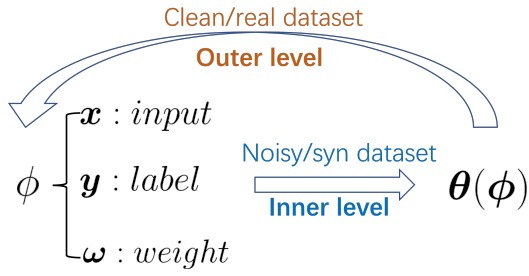

Figure 3: Data-related outer variables.

(f) Several studies (Ren et al., 2018; Hu et al., 2019) have put forth the idea of assigning an instance weight $\phi_i$ to each data point $(\boldsymbol{x}_i, \boldsymbol{y}_i)$ to enhance the model's training process. These instance weights can be considered as outer variables and learned under the guidance of an unbiased validation set. As depicted in Table 2, the methodology of instance reweighting shares similarities with label learning methods and can be encapsulated within a bi-level optimization framework. The initial work by Hu et al. (2019) considers each instance weight as a learnable parameter, which however posed scalability challenges for large datasets. To circumvent this issue, subsequent works (Shu et al., 2019) devise an alternative solution - a weighting network. This network is designed to parameterize instance weights, wherein the input is the instance loss, and the output becomes

Table 3: Same formula with explicit outer level hypergradient.

| Work | Inner Var $\boldsymbol{\theta}$ | Outer Var $\boldsymbol{\phi}$ | Inner Level Problem as Constraint | Outer Level Problem as Main Opt |
|---|---|---|---|---|
| (g) | Follower unit | Leader unit | $\theta^*(\phi) = \arg\min_\phi p(\phi+\theta)\theta - C_f(\theta)$ | $\arg\min_\phi p(\phi+\theta^*(\phi))\phi - C_l(\phi)$ |
| (h) | Model params | Network arch | $\boldsymbol{\theta}^*(\boldsymbol{\phi}) = \arg\min_{\boldsymbol{\theta}} \sum_{(\boldsymbol{x}_i,\boldsymbol{y}_i)\in\mathcal{D}_{train}} l(\boldsymbol{\theta},\boldsymbol{\phi},\boldsymbol{x}_i,\boldsymbol{y}_i)$ | $\arg\min_{\boldsymbol{\phi}} \sum_{(\boldsymbol{x}_j,\boldsymbol{y}_j)\in\mathcal{D}_{val}} l(\boldsymbol{\theta}^*(\boldsymbol{\phi}),\boldsymbol{\phi},\boldsymbol{x}_j,\boldsymbol{y}_j)$ |
| (i) | Model params | Perturbation | $\boldsymbol{\Theta}^*(\boldsymbol{\Phi}) = \arg\min_{\boldsymbol{\Theta}} \mathcal{L}(\boldsymbol{\Theta},\boldsymbol{\Phi})$ | $\arg\min_{\boldsymbol{\Phi}} \mathcal{L}(\boldsymbol{\Theta}(\boldsymbol{\Phi}),\boldsymbol{\Phi})$ |

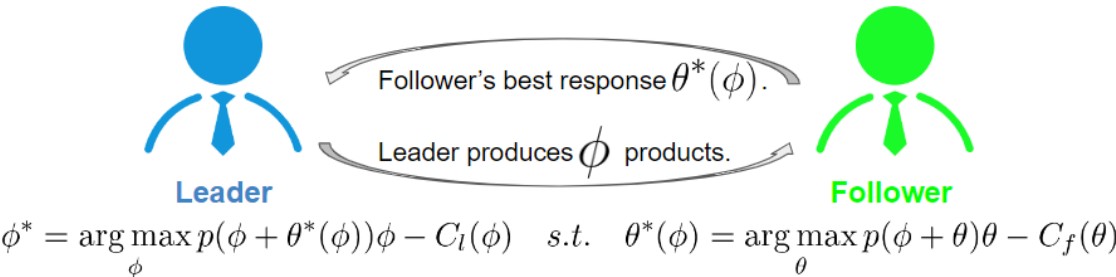

$$\phi^* = \arg\max_\phi p(\phi+\theta^*(\phi))\phi - C_l(\phi) \quad s.t. \quad \theta^*(\phi) = \arg\max_\theta p(\phi+\theta)\theta - C_f(\theta)$$

Figure 4: Stackelberg game as bi-level optimization.

the instance weight, enhancing scalability for large datasets. This innovative approach has been further employed by Xu et al. (2021), who parameterize the atom distance in a molecular structure using a message passing neural network, a mechanism designed to encapsulate graph information effectively. It is crucial to note that efficient optimization is closely tied to the effective parameterization of high-dimensional hyperparameters. Achieving this involves the use of a meticulously designed neural network/input that caters to the demands of the specific problem at hand, ensuring both efficiency and accuracy in the optimization process.

### 3.1.2 Same formula with explicit outer level hypergradient

In instances where the inner and outer levels of the optimization share the same mathematical objective, it is possible for the hypergradient to be derived directly from the outer level. However, such instances are relatively infrequent compared to the previously discussed scenarios. We surmise that this is likely because, in most cases, the outer variable can be directly updated along with the inner variable, either through alternate or joint optimization, without resorting to the more complex approach of bi-level optimization.

(g) In the context of a Stackelberg game (Von Stackelberg, 2010), there are two competing entities - a leading company and a following company, as depicted in Figure 4. These companies compute with each other over the production quantities, where the leader produces $\phi$ units and the follower produces $\theta$ units. The combined price of their output can be expressed as $p(\phi+\theta)$. The cost functions for the leader and the follower are denoted as $C_l(\phi)$ and $C_f(\theta)$ respectively. The resulting profits for the leader and follower can then be calculated as $p(\phi+\theta)\phi - C_l(\phi)$ and $p(\phi+\theta)\theta - C_f(\theta)$. In the Stackelberg game scenario, it is assumed that the leader is aware of the follower's best response. This situation can be aptly framed as a bi-level optimization problem, as detailed in Table 3(g). Importantly, the leader cannot directly optimize $\phi$ at the outer level without taking into account the differentiable constraint imposed at the inner level. This interdependence validates the necessity for the bi-level optimization framework.

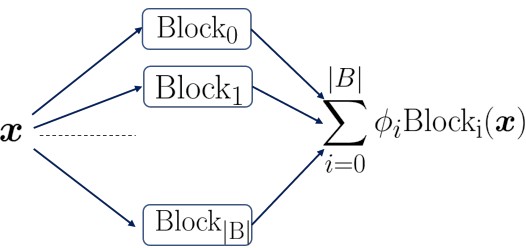

$$\sum_{i=0}^{|B|} \phi_i \mathrm{Block_i}(\boldsymbol{x})$$

(h) The task of searching for an optimal neural network architecture within a defined search space is a critical determinant of the performance of that task. As a neural network architecture can be considered a type of hyperparameter, it appears logical to model the search process as a bi-level optimization problem to enable effective updates, as discussed in Section 3.1.1. However, this task is not straightforward as the network architecture is non-differentiable and

Figure 5: Continuous relaxation of the neural architecture in DARTS.

Table 4: Different formulas without explicit hypergradient.

| Work | Inner Var $\boldsymbol{\theta}$ | Outer Var $\boldsymbol{\phi}$ | Inner Level Problem as Constraint | Outer Level Problem as Main Opt |
|---|---|---|---|---|
| (j) | Conformation | Atom distance | $\boldsymbol{\theta}^*(\boldsymbol{\phi}) = \arg\min_{\boldsymbol{\theta}} \mathcal{L}^{in}(\boldsymbol{\theta}, \boldsymbol{D}_{\boldsymbol{\phi}})$ | $\arg\min_{\boldsymbol{\phi}} \mathcal{L}^{out}(\boldsymbol{\theta}(\boldsymbol{\phi}))$ |
| (k) | Model params | Pretrain hyperparams | $\boldsymbol{\theta}^*(\boldsymbol{\phi}) = \arg\min_{\boldsymbol{\theta}} \mathcal{L}^{in}(\boldsymbol{\theta}, \boldsymbol{\phi}, \mathcal{D}_{prt})$ | $\arg\min_{\boldsymbol{\phi}} \mathcal{L}^{out}(\boldsymbol{\theta}^*(\boldsymbol{\phi}), \boldsymbol{\phi}, \mathcal{D}_{ft})$ |
| (l) | System state | Topology design | $\boldsymbol{\theta}^*(\boldsymbol{\phi}) = \arg\min_{\boldsymbol{\theta}} \mathcal{L}^{in}(\boldsymbol{\theta}, \boldsymbol{\phi})$ | $\arg\min_{\boldsymbol{\phi}} \mathcal{L}^{out}(\boldsymbol{\theta}^*(\boldsymbol{\phi}))$ |
| (m) | Layer output | Model params | $\boldsymbol{\theta} = E(\boldsymbol{\theta}, \boldsymbol{\phi}, \boldsymbol{x})$ | $\arg\min_{\boldsymbol{\phi}} \mathcal{L}^{out}(\boldsymbol{\theta}(\boldsymbol{\phi}), \boldsymbol{x}, \boldsymbol{y})$ |
| (n) | Layer output | Model params | $\dot{\boldsymbol{\theta}}(t) = E(\boldsymbol{\theta}(t), \boldsymbol{\phi}, t), \quad \boldsymbol{\theta}(0) = \boldsymbol{\theta}_0$ | $\arg\min_{\boldsymbol{\phi}} \mathcal{L}^{out}(\boldsymbol{\theta}(t, \boldsymbol{\phi}), \boldsymbol{x}, \boldsymbol{y})$ |

cannot be directly optimized using gradient methods. To address this, Liu et al. (2019) suggest a continuous relaxation of the architecture representation, parameterized by $\boldsymbol{\phi}$ as depicted in Figure 5, and update $\boldsymbol{\phi}$ to pinpoint superior neural network architectures. This scenario can be framed as a bi-level optimization problem as elaborated in Table 3(h). Here, the inner level loss on the training set serves as a constraint and builds a differentiable relationship between the model parameters $\theta$ and the network architectures. At the same time, the outer level loss on the validation set constitutes the main optimization problem, aiming to optimize the network architectures. As there is no ground truth for neural architectures, the outer variable $\boldsymbol{\phi}$ cannot be viewed as a constant as in instance weighting cases (Shu et al., 2019; Chen et al., 2021). Therefore, it is necessary to explicitly update the neural architectures within the context of the outer level loss. It's worth noting that the first-order DARTS method treats $\boldsymbol{\theta}^*(\boldsymbol{\phi})$ as $\boldsymbol{\theta}^*$, independent of $\boldsymbol{\phi}$. In this scenario, bi-level optimization simplifies to alternate optimization. As a result, the performance of the first-order DARTS is typically inferior compared to the original second-order DARTS. It's notable that the conversion of discrete variables to continuous ones holds significant importance in gradient-based bi-level optimization as it offers an efficient method to adjust discrete parameters. This technique finds its application in earlier label learning methodologies, as demonstrated in the works by (Algan & Ulusoy, 2021; Wu et al., 2021). In these cases, discrete one-hot labels are morphed into soft labels for optimization. However, in such instances, the conversion to continuous forms isn't as crucial as in our present context, owing to the existence of alternate methods like instance reweighting for managing label noise.

(i) Dictionary learning methods (Mairal et al., 2010) also belong to this category. These methods aim to find the sparse code $\boldsymbol{\Theta} \in \mathbb{R}^{d \times p}$ and the dictionary $\boldsymbol{\Phi} \in \mathbb{R}^{m \times d}$ to reconstruct the noise measurements $\boldsymbol{Y} \in \mathbb{R}^{m \times p}$. Note that $p$ represents the dataset size, $d$ represents the dictionary size and $m$ represents the feature size of the dictionary feature. The loss function can be written as:

$$\arg\min_{\boldsymbol{\Theta}, \boldsymbol{\Phi}} \mathcal{L}(\boldsymbol{\Theta}, \boldsymbol{\Phi}) = \frac{1}{2}\|\boldsymbol{\Phi}\boldsymbol{\Theta} - \boldsymbol{Y}\|^2 + \gamma\|\boldsymbol{\Theta}\|_1. \tag{6}$$

In this context, $\gamma$ is a regularization parameter. Rather than employing a slow alternate optimization over $\boldsymbol{\Theta}$ and $\boldsymbol{\Phi}$, which ignores their explicit relationship (that a given dictionary $\boldsymbol{\Phi}$ should determine the sparse code $\boldsymbol{\Theta}$), this can be effectively formulated as a bi-level problem, as detailed in Table 3 (i). This approach significantly enhances the rate of convergence.

### 3.1.3 Different formulas without explicit hypergradient

Generally, the inner and outer level objectives share a similar mathematical structure, with the validation set used to gauge the performance of the model parameters. However, there are instances where the inner and outer level objectives differ substantially due to their respective optimization of entirely distinct problems.

(j) The study by Xu et al. (2021) exemplifies such a case. In this research, the prediction of molecular conformation is divided into two levels, each addressing a distinct problem. The inner level problem aims to construct the molecular conformation with the predicted atom distances by leveraging the physical constraint and the outer level problem aims to align the molecular conformation with the ground truth conformations, functioning as the main optimization component. As explicated in Table 4(j), this setup forms a bi-level optimization problem where $\mathcal{L}^{in}$ and $\mathcal{L}^{out}$ denote the reconstruction and alignment losses, respectively. Here, $\boldsymbol{D}_{\boldsymbol{\phi}}$ signifies the predicted atom distances, parameterized by $\boldsymbol{\phi}$, while the inner variable, $\boldsymbol{\theta}$, represents the predicted molecular conformation. The culmination of this process is a finely-tuned neural network capable of accurately predicting atoms' distances.

Table 5: Different formulas without explicit hypergradient.

| Work | Inner Var $\boldsymbol{\theta}$ | Outer Var $\boldsymbol{\phi}$ | Inner Level Problem as Constraint | Outer Level Problem as Main Opt |
|------|------|------|------|------|
| (o) | Model params | Surrogate loss params | $\boldsymbol{\theta}^*(\boldsymbol{\phi}) = \arg\min_{\boldsymbol{\theta}} \mathcal{L}^{in}(\boldsymbol{\theta}, \boldsymbol{\phi}, \mathcal{D}^{train})$ | $\arg\min_{\boldsymbol{\phi}} \mathcal{L}^{out}(\mathcal{L}(\boldsymbol{\theta}, \mathcal{D}^{val})), \mathcal{L}^{in}(\boldsymbol{\theta}^*(\boldsymbol{\phi}), \boldsymbol{\phi}, \mathcal{D}^{val}))$ |
| (p) | Seq params | Struc params | $\boldsymbol{\theta}^*(\boldsymbol{\phi}) = \arg\min_{\boldsymbol{\theta}} \mathcal{L}^{in}(\boldsymbol{\theta}, \boldsymbol{\phi})$ | $\arg\min_{\boldsymbol{\phi}} \mathcal{L}^{out}(\boldsymbol{\theta}^*(\boldsymbol{\phi}), \boldsymbol{\phi})$ |

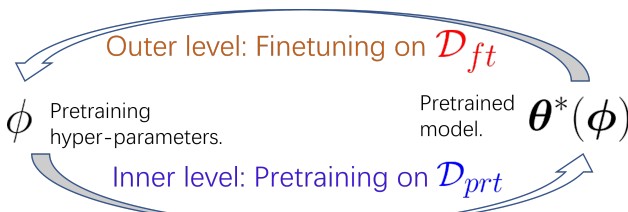

Figure 6: Pretraining and finetuning as bi-level optimization.

(k) Pretraining is a crucial strategy in fields such as computer vision and natural language processing, where identifying the right pretraining hyperparameters can significantly improve the performance on downstream tasks. Raghu et al. (2021) propose an approach to optimize pretraining hyperparameters based on downstream task performance.

This process of pretraining and finetuning can be effectively represented as a bi-level optimization problem, as illustrated in Table 4(k). In this context, $\mathcal{D}_{prt}$ and $\mathcal{D}_{ft}$ denote the pretraining and fine-tuning datasets, respectively. At the inner level, the loss on the pretraining set serves as a constraint and establishes a differentiable connection between the model parameters and the pretraining hyperparameters. The aim at this level is to minimize the pretraining loss. At the outer level, the focus is on optimizing the pretraining hyperparameters by minimizing the loss on the fine-tuning set. This is the main optimization task, and it leverages the differentiable connection established at the inner level. In this way, the pretraining hyperparameters can be effectively tuned to enhance downstream task performances.

(l) The design of a topology that minimizes system cost is an important problem in science (Christiansen et al., 2001; Zehnder et al., 2021). In the context of topology design, the system reaches an equilibrium state, denoted by $\boldsymbol{\theta}^*(\boldsymbol{\phi})$, once a topology $\boldsymbol{\phi}$ is provided. This equilibrium state is achieved by minimizing the energy function $\mathcal{L}^{in}(\boldsymbol{\theta}, \boldsymbol{\phi})$. Following this, the system cost can be calculated as $\mathcal{L}^{out}(\boldsymbol{\theta}^*(\boldsymbol{\phi}))$. This process can be articulated as a bi-level optimization problem, as detailed in Table 4 (l). The incorporation of an energy constraint also proves valuable in better simulating soft-body physics (Rojas et al., 2021).

This concept extends to implicit layers, which encompass (m) Deep Equilibrium Models (DEQ) (Bai et al., 2019) and (n) Neural Ordinary Differential Equations (NeuralODE) (Chen et al., 2018). Both models entail an equation constraint. DEQ enhances the effectiveness of neural network (NN) representation $\boldsymbol{\theta}$ with input $\boldsymbol{x}$ by employing an infinite-depth layer $E$. This is expressed as $\boldsymbol{\theta} = E(\boldsymbol{\theta}, \boldsymbol{\phi}, \boldsymbol{x})$, which establishes a relationship between the equilibrium point $\boldsymbol{\theta}$ and the model parameters $\boldsymbol{\phi}$. Subsequently, the supervised loss is utilized to update the model parameters $\phi$ via this relationship, thus forming a bi-level optimization problem, as described in Table 4 (m). NeuralODE follows a similar structure as displayed in Table 4 (n). The sole difference resides in the equation constraint applied in this context, which is an Ordinary Differential Equation: $\dot{\boldsymbol{\theta}}(t) = E(\boldsymbol{\theta}(t), \boldsymbol{\phi}, t)$.

### 3.1.4 Different formulas with explicit hypergradient

In situations where the inner and outer level objectives are distinct from each other, some scenarios feature a tangible hypergradient. Such cases include (o) learning the surrogate loss function (Grabocka et al., 2019) and (p) protein representation learning (Chen et al., 2022b), among others.

(o) In machine learning, proxies of misclassification rate such as cross-entropy are often used to approximate actual losses, primarily due to their non-differentiable and discontinuous nature. To bridge this gap, a study by Grabocka et al. (2019) introduces a surrogate neural network for accurate approximation of these true losses. This process involves a bi-level optimization formulation, as outlined in Table 5(o). The inner level focuses on minimizing the surrogate loss $\mathcal{L}_{in}$ by optimizing model parameters $\boldsymbol{\theta}$, while the outer level refines the surrogate loss $\boldsymbol{\phi}$ to resemble the true loss $\mathcal{L}$. The main objective here is to minimize the distance between the true loss $\mathcal{L}$ and the surrogate loss $\mathcal{L}^{in}$ through the outer level optimization process. This

method facilitates the effective and virtual minimization of any non-differentiable and non-decomposable loss function, such as the misclassification rate.

(p) Protein pretraining plays a pivotal role in facilitating downstream tasks, and incorporating bio-chemical constraints into the learning process can enhance its performance. In this context, the protein modeling neural network deals with two types of information: sequential representation parameterized by $\boldsymbol{\theta}$ and structural representation parameterized by $\boldsymbol{\phi}$. The study by Chen et al. (2022b) utilizes the bio-chemical constraint that every protein sequence is associated with a particular protein structure, and formulates protein pretraining as a bi-level optimization problem, as detailed in Table 5(p). As seen, the inner level establishes the connection between the sequential and structural information by minimizing the negative mutual information loss, $\mathcal{L}^{in}$, acting as the biochemical constraint. Simultaneously, the outer level refines the structural parameters by minimizing the pretraining loss, $\mathcal{L}^{out}$, serving as the main optimization component. Overall, this pretraining scheme bolsters the performance of protein representation learning.

### 3.2 Multi-task formulation

Multi-task formulation, in contrast to single-task formulation, seeks to extract meta-knowledge spanning across various tasks. While the choice of hyperparameters in a single-task formulation is broad and diverse as we illustrate in Section 3.1, in the context of multi-task formulation, the choice of meta-knowledge tends to be more constrained, making its formulation relatively straightforward. Primarily, two types of meta-knowledge are embodied by the outer variable in multi-task formulation: model initialization and optimizer, which we will elaborate upon in the following subsections.

#### 3.2.1 Model initialization

The first kind of meta-knowledge is model initialization, which is useful in the data-scarce scenario (Hiller et al., 2022; Liu et al., 2020; Li et al., 2017). The work (Finn et al., 2017) proposes Model-Agnostic Meta-Learning (MAML) to train the parameters of the model across a family of tasks to generate a good model initialization. A good model initialization means a few steps on this initialization reach a good solution. Given a model initialization $\boldsymbol{\phi}$, the model parameters fine-tuned on the task $i$ after a gradient descent step can be written as:

$$\boldsymbol{\theta}_i(\boldsymbol{\phi}) = \boldsymbol{\phi} - \eta \frac{\partial \mathcal{L}_i^{in}(\boldsymbol{\phi}, \mathcal{D}_i^{train})}{\partial \boldsymbol{\phi}^\top}. \tag{7}$$

The updated model parameters are expected to perform well on the validation set:

$$\boldsymbol{\phi}^* = \arg\min_{\boldsymbol{\phi}} \sum_{i=1}^{M} \mathcal{L}^{out}(\boldsymbol{\theta}_i(\boldsymbol{\phi}), \mathcal{D}_i^{val}). \tag{8}$$

$$\text{s.t.} \quad \boldsymbol{\theta}_i(\boldsymbol{\phi}) = \arg\min_{\boldsymbol{\theta}} \mathcal{L}_i^{in}(\boldsymbol{\phi}, \mathcal{D}_i^{train}). \tag{9}$$

This forms a bi-level optimization problem as shown in Figure 7. MAML is applicable to a wide range of learning tasks including classification, regression, etc.

A single initialization may not be able to generalize to all tasks and thus some research work propose to learn different initialization for different tasks. The method (Vuorio et al., 2019) modulates the initialization according to the task mode and adapts quickly by gradient updates. The approach (Yao et al., 2019) clusters tasks hierarchically and adapts the initialization according to the cluster. According to the task formulation, maybe only part of the model parameters need to be updated, which can save much memory considering millions of parameters for the model. The work (Lee et al., 2019) fixes the user embedding and item embedding and only updates the interaction parameters in the meta-learned phase. The method (Rusu et al., 2018) proposes to map the high dimensional parameter space to a low-dimensional latent where they can perform MAML.

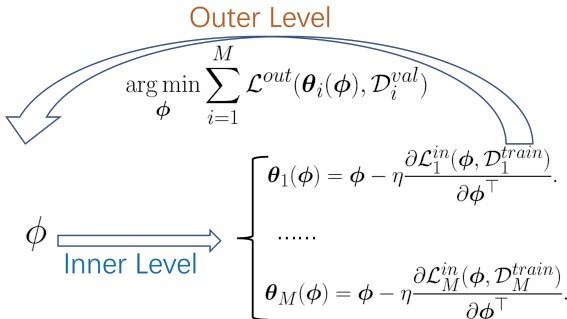

Figure 7: Illustration of MAML.

Besides, some analyses (Zou et al., 2021) are also proposed for choosing the inner loop learning rate. Last but not least, domain knowledge such as biological prior (Yao et al., 2021) can also be incorporated into the MAML modeling where they propose a region localization network to customize the initialization to each assay. Tack et al. (2022) adapt the temporal ensemble of the meta-learner to generate the target model. Hiller et al. (2022) develop a novel method to increase adaptation speed inspired by preconditioning. Guan et al. (2022) analyze modern meta-learning algorithms and give a detailed analysis of stability.

### 3.2.2 Optimizer

Another kind of meta-knowledge across tasks is an optimizer. Previous optimizers such as Adam are designed by hand and may be sub-optimal. The work (Andrychowicz et al., 2016) proposes to learn an optimizer for a family of tasks. As shown in Figure 8, the learned optimizer is parameterized by $\phi$ where an LSTM takes the state as input and outputs the update. In this way, for the $i_{th}$ task, the model parameters $\theta$ are connected with the optimizer parameters via:

$$\theta_{t+1}^i(\phi) = \theta_t^i + g_t^i(\phi). \tag{10}$$

$$[g_t^i(\phi), h_{t+1}^i] = \mathrm{OPT}(\nabla_{\theta_t^i} \mathrm{l_t}, \mathrm{h_t^i}, \phi). \tag{11}$$

Here OPT denotes the learned LSTM optimizer and $h$ represents the hidden state. The optimizer is updated to improve the validation performance over a horizon T, and this can be written as:

$$\phi^* = \arg\min_\phi \sum_{t=1}^T \mathcal{L}_{out}(\theta_t^i(\phi), \mathcal{D}_{val}^i). \tag{12}$$

Overall, this can be formulated as a bi-level optimization problem where the inner level builds the connection between the model parameters $\theta$ and the optimizer parameters $\phi$ by minimizing the training loss, and the outer level updates the optimizer by minimizing the validation loss. Formally, the formulation of bi-level optimization across a family of M tasks can be written as:

$$\phi^* = \arg\min_\phi \sum_{i=1}^M \sum_{t=1}^\top \mathcal{L}_{out}(\theta_t^i(\phi), \mathcal{D}_{val}^i). \tag{13}$$

$$\text{s.t.} \quad \theta_{t+1}^i(\phi) = \theta_t^i + g_t^i(\phi); [g_t^i(\phi), h_{t+1}^i] = \mathrm{OPT}(\nabla_{\theta_t^i} \mathrm{l_t}, \mathrm{h_t^i}, \phi). \tag{14}$$

To optimize millions of parameters, the LSTM is designed to be coordinatewise, which means every parameter shares the same LSTM. This greatly alleviates the computational burden. Besides, some preprocessing and postprocessing techniques are proposed to rescale the inputs and the outputs of LSTM into a normal range. One key challenge of learning to optimize is the generalization to longer horizons or unseen optimizees and many research works try to mitigate this challenge. The work (Metz et al., 2019) proposes to use an MLP layer instead of an LSTM to parameterize the optimizer and smooth the loss scope by dynamic gradient reweighting.

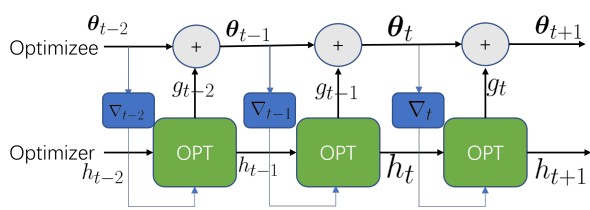

Figure 8: LSTM optimizer.

The approach (Wichrowska et al., 2017) proposes a hierarchical RNN formed by three RNN layers that could communicate from bottom to up and this hierarchical design achieves better generalization. The work (Lv et al., 2017) proposes training tricks such as random scaling to improve the generalization. Besides the above, some work (Knyazev et al., 2021)(Kang et al., 2021) proposes to directly predict parameters, which can be seen as a specially learned optimizer without any gradient update.

Recent works also try to incorporate the existing optimizer into the optimizer learning, which can leverage both the existing prior and the learning capacity. The key is to replace the constant (i.e. scalar/vector/matrix) in the existing optimizer with learnable parameters. HyperAdam (Wang et al., 2019) learns the combination weights and decay rates according to the task. The method (Gregor & LeCun, 2010) first writes the ISTA as a recurrent formula and then parameterizes the coefficients as the outer variable. The approach (Shu et al., 2020) designs a Meta-LR-Schedule-Net which takes the loss value and the state as input and outputs the learning rate for the current iteration. The work (Ravi & Larochelle, 2016) proposes to parameterize the weight coefficient and the learning rate.

Besides the model initialization and the parameterized optimizer, there is some other meta-knowledge like the loss function learning (Gao et al., 2022). They propose to learn a generic loss function to train a robust DNN model that can perform well on out-of-distribution tasks. Given a parametric loss function as the outer variable $\phi$, the inner level yields the optimized model parameters $\theta(\phi)$ by minimizing the training loss on the source domain. Then a good parametric loss can be identified by minimizing the validation loss on the target domain. The above process formulates a bi-level optimization problem, which is given by

$$\phi^* = \arg\min_{\phi} \sum_{i=1}^{N} \mathcal{L}^{out}(\theta^*(\phi), \mathcal{D}_{val}^i). \tag{15}$$

$$\text{s.t.} \quad \theta^*(\phi) = \arg\min_{\theta} \sum_{i=1}^{M} \mathcal{L}^{in}(\theta, \phi, \mathcal{D}_{train}^i). \tag{16}$$

Here $\mathcal{L}^{out}$ is a loss function to measure the performance on target domains and $N$, and $M$ represent the number of target domain and source domain tasks, respectively. They further propose to compute the hyper-gradient by leveraging the implicit function theorem.

## 4 Optimization

Gradient-based bi-level optimization requires the hypergradient computation of $\frac{d\mathcal{L}^{out}}{d\phi}$ in the outer level. The hypergradient $\frac{d\mathcal{L}^{out}}{d\phi}$ can be unrolled via the chain rule as:

$$\frac{d\mathcal{L}^{out}}{d\phi} = \frac{\partial \mathcal{L}^{out}}{\partial \theta} \frac{\partial \theta(\phi)}{\partial \phi} + \frac{\partial \mathcal{L}^{out}}{\partial \phi}. \tag{17}$$

Here $\frac{\partial \theta(\phi)}{\partial \phi}$ often involves second-order gradient computation and thus are resource demanding. There are generally four types of methods to calculate $\frac{\partial \theta(\phi)}{\partial \phi}$: explicit gradient update in Section 4.1, explicit proxy update in Section 4.2, implicit function update in Section 4.3, and closed-form method in Section 4.4, where the previous three are approximation methods for general functions with the difference in how to build the connection between $\theta$ and $\phi$ and the last one is an accurate method for certain functions. Subsequent to these detailed descriptions, a comprehensive analysis comparing their time and space complexities will be carried out in Section 4.5.

### 4.1 Explicit gradient update

The explicit gradient update is the most straight-forward one which approximates $\theta$ via some optimizer directly:

$$\theta_t = \text{OPT}(\theta_{t-1}, \phi), \quad t = 1, \cdots, T. \tag{18}$$

Here $T$ denotes the number of iterations, $\theta$ represents the model parameters and other optimization variables like momentum, OPT represents the optimization algorithm like SGD, and $\phi$ denotes the outer variable in

the training process. Note that when OPT is the SGD optimizer and only one gradient descent step is considered, Eq. (18) becomes

$$\boldsymbol{\theta}(\boldsymbol{\phi}) = \boldsymbol{\theta} - \eta \frac{\partial \mathcal{L}^{in}(\boldsymbol{\theta}, \boldsymbol{\phi}, \mathcal{D}^{train})}{\partial \boldsymbol{\theta}^{\top}}. \tag{19}$$

In this case, we can compute the hypergradient as:

$$\frac{\partial \boldsymbol{\theta}(\boldsymbol{\phi})}{\partial \boldsymbol{\phi}} = -\eta \frac{\partial^2 \mathcal{L}^{in}(\boldsymbol{\theta}, \boldsymbol{\phi}, \mathcal{D}^{train})}{\partial \boldsymbol{\theta}^{\top} \partial \boldsymbol{\phi}}. \tag{20}$$

This process often requires the second-order gradient computation. In some cases, the first-order approximation can be adopted to replace the second-order gradient in (Liu et al., 2019; Finn et al., 2017; Nichol et al., 2018). Besides, Liu et al. (2019) uses the finite difference approximation technique to compute the second-order gradient efficiently.

$$\frac{\partial^2 \mathcal{L}^{in}(\boldsymbol{\theta}, \boldsymbol{\phi}, \mathcal{D}^{train})}{\partial \boldsymbol{\theta}^{\top} \partial \boldsymbol{\phi}} \frac{\partial \mathcal{L}^{out}}{\partial \boldsymbol{\theta}} \approx \frac{\frac{\partial \mathcal{L}^{in}(\boldsymbol{\theta}^{+}, \boldsymbol{\phi}, \mathcal{D}^{train})}{\partial \boldsymbol{\phi}} - \frac{\partial \mathcal{L}^{in}(\boldsymbol{\theta}^{-}, \boldsymbol{\phi}, \mathcal{D}^{train})}{\partial \boldsymbol{\phi}}}{2\epsilon}, \tag{21}$$

where $\boldsymbol{\theta}^{\pm} = \boldsymbol{\theta} \pm \epsilon \frac{\partial \mathcal{L}^{out}(\boldsymbol{\theta}, \boldsymbol{\phi})}{\partial \boldsymbol{\theta}}$. This avoids the expensive computational cost of the Hessian matrix. Furthermore, the work (Deleu et al., 2022) proposes to adopt infinitely small gradient steps to solve the inner level task, which leads to a continuous-time bi-level optimization solver:

$$\frac{d\boldsymbol{\theta}(t)}{dt} = \frac{\partial \mathcal{L}^{in}(\boldsymbol{\theta}, \boldsymbol{\phi}, \mathcal{D}^{train})}{\partial \boldsymbol{\theta}^{\top}}. \tag{22}$$

In this way, the final output is the solution of an ODE. One great advantage of this formulation is making the fixed and discrete number of gradient steps the length of the trajectory, which serves as a continuous variable and is also learnable. This work also proposes to use forward mode differentiation to compute the hypergradient where the memory does not scale with the length of the trajectory. A similar continuous bi-level solver is used in Yuan & Wu (2021).

Generally speaking, the update is not limited to one step nor SGD optimizer, which makes the hypergradient computation process complicated. There are generally two modes (Franceschi et al., 2017) to compute the hypergradient: forward mode and reverse mode.

**Forward mode.** Forward mode methods apply the chain rule to the composite functions:

$$\frac{d\boldsymbol{\theta}_t}{d\boldsymbol{\phi}} = \frac{\partial \mathrm{OPT}(\boldsymbol{\theta}_{t-1}, \boldsymbol{\phi})}{\partial \boldsymbol{\theta}_{t-1}} \frac{d\boldsymbol{\theta}_{t-1}}{d\boldsymbol{\phi}} + \frac{\partial \mathrm{OPT}(\boldsymbol{\theta}_{t-1}, \boldsymbol{\phi})}{\partial \boldsymbol{\phi}}. \tag{23}$$

Then the matrics are defined as:

$$\boldsymbol{Z}_t = \frac{d\boldsymbol{\theta}_t}{d\boldsymbol{\phi}}, \quad \boldsymbol{A}_t = \frac{\partial \mathrm{OPT}(\boldsymbol{\theta}_t, \boldsymbol{\phi})}{\partial \boldsymbol{\theta}_t}, \quad \boldsymbol{B}_t = \frac{\partial \mathrm{OPT}(\boldsymbol{\theta}_t, \boldsymbol{\phi})}{\partial \boldsymbol{\phi}_t}. \tag{24}$$

Thus the Eq.(23) can be written as:

$$\boldsymbol{Z}_t = \boldsymbol{A}_t \boldsymbol{Z}_{t-1} + \boldsymbol{B}_{t-1}. \tag{25}$$

In this way, $\boldsymbol{Z}_T$ can be written as:

$$\boldsymbol{Z}_T = \boldsymbol{A}_T \boldsymbol{Z}_{T-1} + \boldsymbol{B}_{T-1} \tag{26}$$

$$= \sum_{t=1}^{T} (\prod_{s=t+1}^{T} \boldsymbol{A}_s) \boldsymbol{B}_t. \tag{27}$$

which yields the final hypergradient.

**Reverse mode.** The reverse mode approach originates from Lagrangian optimization. The Lagrangian of bi-level problems can be formulated as:

$$\mathcal{L}(\boldsymbol{\theta}, \boldsymbol{\phi}, \boldsymbol{\gamma}) = \mathcal{L}^{out}(\boldsymbol{\theta}_T) + \sum_{t=1}^{T} \boldsymbol{\gamma}_t (\mathrm{OPT}(\boldsymbol{\theta}_{t-1}, \boldsymbol{\phi}) - \boldsymbol{\theta}_t). \tag{28}$$

The main calculation of partial derivatives can be presented as follows:

$$\frac{\partial \mathcal{L}(\boldsymbol{\theta}, \boldsymbol{\phi}, \boldsymbol{\gamma})}{\partial \boldsymbol{\theta}_t} = \boldsymbol{\gamma}_{t+1} \boldsymbol{A}_{t+1} - \boldsymbol{\gamma}_t, \quad t \in \{1, \cdots, T-1\}. \tag{29}$$

$$\frac{\partial \mathcal{L}(\boldsymbol{\theta}, \boldsymbol{\phi}, \boldsymbol{\gamma})}{\partial \boldsymbol{\theta}_T} = \frac{\partial \mathcal{L}^{out}}{\partial \boldsymbol{\theta}_T} - \boldsymbol{\gamma}_T. \tag{30}$$

$$\frac{\partial \mathcal{L}(\boldsymbol{\theta}, \boldsymbol{\phi}, \boldsymbol{\gamma})}{\partial \boldsymbol{\phi}} = \sum_{t=1}^{T} \boldsymbol{\gamma}_t \boldsymbol{B}_t. \tag{31}$$

Upon setting the partial derivatives in Eq.(29) and Eq.(30) to zero, we can deduce the values of $\boldsymbol{\gamma}_t$. Incorporating this solution with Eq. (31), we find:

$$\frac{\partial \mathcal{L}(\boldsymbol{\theta}, \boldsymbol{\phi}, \boldsymbol{\gamma})}{\partial \boldsymbol{\phi}} = \frac{\partial \mathcal{L}^{out}}{\partial \boldsymbol{\theta}_T} \sum_{t=1}^{T} (\prod_{s=t+1}^{T} \boldsymbol{A}_s) \boldsymbol{B}_t. \tag{32}$$

It can be seen that the reverse mode produces the same solution as the forward mode. Works by Shaban et al. (2019); Luketina et al. (2016) suggest disregarding long-term dependencies to increase efficiency.

## 4.2 Explicit proxy update

Besides using explicit gradient update to solve the inner level task, a more direct way (MacKay et al., 2019; Bae & Grosse, 2020; Lorraine & Duvenaud, 2018) is to fit a proxy network $P_\alpha(\cdot)$ which takes the outer variable as input and outputs the inner variable,

$$\boldsymbol{\theta}^* = P_\alpha(\boldsymbol{\phi}). \tag{33}$$

There are two ways to train the proxy: global and local. The global way aims to learn a proxy for all $\boldsymbol{\phi}$ by minimizing $\mathcal{L}^{in}(P_\alpha(\boldsymbol{\phi}), \boldsymbol{\phi}, \mathcal{D}^{train})$ for all $\boldsymbol{\phi}$ against $\boldsymbol{\alpha}$ while the local way minimizes $\mathcal{L}^{in}(P_\alpha(\boldsymbol{\phi}), \boldsymbol{\phi}, \mathcal{D}^{train})$ against $\boldsymbol{\alpha}$ for a neighborhood of $\boldsymbol{\phi}$.

A special case (Bohdal et al., 2021) is to design the proxy as a weighted average of the perturbed inner variable. This work adopts an evolutionary algorithm to obtain an approximate solution for $\boldsymbol{\theta}$. By perturbing $\boldsymbol{\theta}$ to $\boldsymbol{\theta}_k$ for $K$ times, they compute the training losses as $\{l_k(\boldsymbol{\phi})\}_{k=1}^{K}$ where $l_k(\boldsymbol{\phi}) = \mathcal{L}^{in}(\boldsymbol{\theta}_k, \boldsymbol{\phi}, \mathcal{D}^{train})$. Then the weights for each perturbed loss are,

$$\omega_1, \omega_2, \cdots, \omega_K = \text{softmax}([-l_1(\boldsymbol{\phi}), -l_2(\boldsymbol{\phi}), \cdots, -l_K(\boldsymbol{\phi})]/\tau), \tag{34}$$

where $\tau > 0$ is a hyperparameter. Last, the proxy is obtained as

$$\boldsymbol{\theta}^* = \omega_1 \boldsymbol{\theta}_1 + \omega_2 \boldsymbol{\theta}_1 + \cdots + \omega_K \boldsymbol{\theta}_K. \tag{35}$$

Compared with explicit gradient update methods, these proxy methods can adopt deep learning modules to directly build the relationship between the inner variable and the outer variable. Thus these methods generally require less memory while are less accurate due to the rough approximation brought by the deep learning module.

## 4.3 Implicit function update

The hypergradient computation of the explicit gradient update methods relies on the path taken at the inner level, while the implicit function update makes use of the implicit function theorem to derive a more accurate hypergradient without vanishing gradients or memory constraints issues. First, the derivate of the inner level is set as zero:

$$\frac{\partial \mathcal{L}(\boldsymbol{\theta}, \boldsymbol{\phi})}{\partial \boldsymbol{\theta}^\top} = \mathbf{0}. \tag{36}$$

Then according to the implicit function theorem, we have:

$$\frac{\partial^2 \mathcal{L}(\boldsymbol{\theta},\boldsymbol{\phi})}{\partial \boldsymbol{\theta}^\top \partial \boldsymbol{\theta}} \frac{\partial \boldsymbol{\theta}(\boldsymbol{\phi})}{\partial \boldsymbol{\phi}} + \frac{\partial \mathcal{L}^2(\boldsymbol{\theta},\boldsymbol{\phi})}{\partial \boldsymbol{\theta}^\top \partial \boldsymbol{\phi}} = \boldsymbol{0}. \tag{37}$$

At last, we can compute the hypergradient as:

$$\frac{\partial \boldsymbol{\theta}(\boldsymbol{\phi})}{\partial \boldsymbol{\phi}} = -\left(\frac{\partial^2 \mathcal{L}(\boldsymbol{\theta}(\boldsymbol{\phi}),\boldsymbol{\phi})}{\partial \boldsymbol{\theta}^\top \partial \boldsymbol{\theta}}\right)^{-1} \frac{\partial \mathcal{L}^2(\boldsymbol{\theta},\boldsymbol{\phi})}{\partial \boldsymbol{\theta}^\top \partial \boldsymbol{\phi}}. \tag{38}$$

Take iMAML (Rajeswaran et al., 2019) as an example. To keep the dependency between the model parameters $\boldsymbol{\theta}$ and the meta parameters $\boldsymbol{\phi}$, iMAML proposes a constraint on the inner level task:

$$\boldsymbol{\theta}_i(\boldsymbol{\phi}) = \arg\min_{\boldsymbol{\theta}} \mathcal{L}_i^{in}(\boldsymbol{\theta}, \mathcal{D}_i^{train}) + \frac{\lambda}{2}\|\boldsymbol{\phi} - \boldsymbol{\theta}\|^2. \tag{39}$$

The regularization strength $\lambda$ controls the strength of the prior $\boldsymbol{\phi}$ relative to the dataset. The iMAML bi-level optimization task can be formulated as:

$$\boldsymbol{\phi}^* = \arg\min_{\boldsymbol{\phi}} \sum_{i=1}^{M} \mathcal{L}^{out}(\boldsymbol{\theta}_i(\boldsymbol{\phi}), \mathcal{D}_i^{val}). \tag{40}$$

$$\text{s.t.} \quad \boldsymbol{\theta}_i(\boldsymbol{\phi}) = \arg\min_{\boldsymbol{\theta}} \mathcal{L}_i^{in}(\boldsymbol{\theta}, \mathcal{D}_i^{train}) + \frac{\lambda}{2}\|\boldsymbol{\phi} - \boldsymbol{\theta}\|^2, i = 1, \cdots, M. \tag{41}$$

The hyper-gradient can be computed as:

$$\frac{d\boldsymbol{\theta}_i(\boldsymbol{\phi})}{d\boldsymbol{\phi}} = \left(\mathbf{I} + \frac{1}{\lambda}\frac{\partial^2 \mathcal{L}^{in}(\boldsymbol{\theta}_i, \mathcal{D}^{train})}{\partial \boldsymbol{\theta}_i^\top \partial \boldsymbol{\theta}_i}\right)^{-1}, \tag{42}$$

which is independent of the inner level optimization path. In this case, the hypergradient can be computed as the solution $\boldsymbol{g}$ to a linear system $\frac{\partial^2 \mathcal{L}^{in}(\boldsymbol{\theta}, \mathcal{D}^{train})}{\partial \boldsymbol{\theta}^\top \partial \boldsymbol{\theta}}\boldsymbol{g} = \frac{\partial \mathcal{L}^{out}}{\partial \boldsymbol{\theta}}$. More specifically, $\boldsymbol{g}$ can be seen as the approximate solution to the following optimization problem:

$$\arg\min_{\boldsymbol{\omega}} \boldsymbol{\omega}^\top \left(\boldsymbol{I} + \frac{1}{\lambda}\frac{\partial^2 \mathcal{L}^{in}(\boldsymbol{\theta})}{\partial \boldsymbol{\theta}^\top \partial \boldsymbol{\theta}}\right)\boldsymbol{\omega} - \boldsymbol{\omega}^\top \frac{\partial \mathcal{L}^{out}(\boldsymbol{\theta})}{\partial \boldsymbol{\theta}^\top}. \tag{43}$$

Conjugate gradient methods can be applied to solve this problem where only Hessian-vector products are computed and the hessian matrix is not explicitly formed. This efficient algorithm is also used in HOAG (Pedregosa, 2016) to compute the hypergradient. Besides the linear system way to compute hypergradient, the work in Lorraine et al. (2020) proposes to unroll the above term into the Neumann Series:

$$\left(\frac{\partial^2 \mathcal{L}^{in}(\boldsymbol{\theta})}{\partial \boldsymbol{\theta}^\top \partial \boldsymbol{\theta}}\right)^{-1} = \sum_{i=0}^{\inf} \left(\boldsymbol{I} - \frac{\partial^2 \mathcal{L}^{in}(\boldsymbol{\theta})}{\partial \boldsymbol{\theta}^\top \partial \boldsymbol{\theta}}\right)^i. \tag{44}$$

The first $i_{th}$ steps' result are used to approximate the computation if $\boldsymbol{I} - \frac{\partial^2 \mathcal{L}(\boldsymbol{\theta}(\boldsymbol{\phi}),\boldsymbol{\phi})}{\partial \boldsymbol{\theta}^\top \partial \boldsymbol{\theta}}$ is contractive. This can avoid the expensive computation of the inverse Hessian.

## 4.4 Closed-form update

While the above three methods provide an approximate solution for general loss functions, we here consider deriving a closed-form connection between $\boldsymbol{\theta}$ and $\boldsymbol{\phi}$ from

$$\boldsymbol{\theta}(\boldsymbol{\phi}) = \arg\min_{\boldsymbol{\theta}} \mathcal{L}^{in}(\boldsymbol{\theta}, \boldsymbol{\phi}, \mathcal{D}^{train}), \tag{45}$$

which is only applicable for some special cases. Bertinetto et al. (2019) propose ridge regression as part of its internal model for closed-form solutions. Assume a linear predictor $f$ parameterized by $\boldsymbol{\theta}$ is considered

as the final layer of a CNN parameterized by $\phi$. Asume the input $\boldsymbol{X} \in \mathbb{R}^{n \times p}$ and the output $\boldsymbol{Y} \in \mathbb{R}^{n \times o}$ where $n$ represents the number of data points and $d, o$ represents the input dimension and the output dimension, respectively. Denote the CNN as $\phi(\boldsymbol{X})$: $\mathbb{R}^p \to \mathbb{R}^e$ and then the predictor's output is $\phi(\boldsymbol{X})\boldsymbol{\theta}$ where $\phi(\boldsymbol{X}) \in \mathbb{R}^{n \times e}$ and $\boldsymbol{\theta} \in \mathbb{R}^{e \times o}$. The $i_{th}$ inner level optimization task can be written as:

$$\boldsymbol{\theta}_i(\phi) = \arg\min_{\boldsymbol{\theta}} \|\phi(\boldsymbol{X}_i)\boldsymbol{\theta} - \boldsymbol{Y}_i\|^2 + \lambda\|\boldsymbol{\theta}\|^2. \tag{46}$$

where $\lambda$ controls the strength of $L^2$ regularization. The closed form solution is $\boldsymbol{\theta}_i(\phi) = \phi(\boldsymbol{X}_i)^\top(\phi(\boldsymbol{X}_i)\phi(\boldsymbol{X}_i)^\top + \lambda\mathbf{I})^{-1}\boldsymbol{Y}_i$. To sum up, to extract meta-knowledge $\phi$ in the feature extractor, we have the bi-level optimization formulation as:

$$\phi^* = \arg\min_{\phi} \sum_{i=1}^{M} \mathcal{L}^{out}(\boldsymbol{\theta}_i(\phi), \mathcal{D}_i^{val}). \tag{47}$$

$$\text{s.t.} \quad \boldsymbol{\theta}_i(\phi) = \phi(\boldsymbol{X}_i)^\top(\phi(\boldsymbol{X}_i)\phi(\boldsymbol{X}_i)^\top + \lambda\mathbf{I})^{-1}\boldsymbol{Y}_i, i = 1, \cdots, M. \tag{48}$$

Applying Newton's method to logistic regression, yields a series of weighted least squares (or ridge regression) problems. This is also a closed-form solution but requires a few steps.

Another special case is to assume the model to be wide enough. Recent work (Jacot et al., 2018; Lee et al., 2017) build the correspondence between the NNGP kernel and the Bayesian Neural Network and the correspondence between the NTK kernel and the gradient trained Neural Network with MSE loss. In this case, the inner level can have a closed-form solution. The works (Nguyen et al., 2020; Yuan & Wu, 2021) treat the data as the outer variable for data distillation and adversarial attack tasks respectively, which yield better-distilled samples and adversarial attacks. These algorithms can be achieved by NTK tool (Novak et al., 2019) easily. The approach (Dukler et al., 2021) treats the instance weights as the outer variable and assumes the pretrained model with linear representation to yield a closed-form solution for the inner level task. Besides assuming the inner level as ridge regression and least squares, some works (Ghadimi & Wang, 2018; Yang et al., 2021) also assume the inner level loss function is strongly convex and propose effective algorithms to better solve the bi-level optimization problem.

### 4.5 Comparative Analysis.

This subsection delves into a comparative analysis of the time and space complexity involved in computing the hypergradient $\frac{d\mathcal{L}^{out}}{d\phi}$ in Eq.(17). We will evaluate the following four methods for this purpose, with a summary provided in Table 6 for reference.

**Explicit gradient update.** Denote the time of computing Eq.(18) as $c(d, m)$. Following the approach (Franceschi et al., 2017), the product between the Jacobian matrix in Eq.(18) and any vector can be computed within a time complexity of $\mathcal{O}(c(d, m))$. In the context of the forward mode for explicit gradient update (termed *explicit gradient forward*), the time complexity becomes $\mathcal{O}(Tmc(d, m))$. This is because it requires $T$ iterations, and each iteration involves $m$ Jacobian-vector products. The space complexity for the forward mode is $\mathcal{O}(d + m)$, as the inner variable $\boldsymbol{\theta}$ can be overwritten in each iteration. Contrastingly, for the reverse mode (termed *explicit gradient reverse*), the time complexity is $\mathcal{O}(Tc(d, m))$ since it involves only one Jacobian-vector product per iteration. However, the space complexity increases to $\mathcal{O}(Td + m)$, as the inner variable cannot be overwritten. In summary, the explicit gradient forward and explicit gradient reverse methods offer a trade-off between time and space complexities: the former minimizes memory usage at the cost of increased computation time, whereas the latter accelerates computation at the expense of memory.

**Explicit proxy update.** The implementation of this method necessitates the training and utilization of a proxy network. The space complexity is $\mathcal{O}(d + m)$ as it maintains both the inner variable and the outer variable. Let $t(d, m)$ represent the time required to train the proxy network, and $i(d, m)$ denote the inference time. The time complexity is $\mathcal{O}(t(d, m) + i(d, m))$. While it's challenging to draw direct comparisons with other methods due to the potential variations in proxy network designs, this method is generally considered to be both time- and space-efficient. However, this comes at the expense of solution accuracy. Often, the explicit proxy update method may not perform as well as other methods because the proxy network $\boldsymbol{\theta}^* = P_\alpha(\phi)$ may not accurately approximate the relation between the inner variable and the outer variable.

Table 6: Comparison on time and space complexity.

| Method | Time | Space |
|---|---|---|
| Explicit gradient forward | $\mathcal{O}(Tmc(d,m))$ | $\mathcal{O}(d+m)$ |
| Explicit gradient reverse | $\mathcal{O}(Tc(d,m))$ | $\mathcal{O}(Td+m)$ |
| Explicit proxy update | $t(d,m)+i(d,m)$ | $\mathcal{O}(d+m)$ |
| Implicit function update | $\mathcal{O}((Km+T)c(d,m))$ | $\mathcal{O}(d+m)$ |
| Closed-form update | $\mathcal{O}(N^3)$ | $\mathcal{O}(N^2)$ |

**Implicit function update.** The implicit function update method capitalizes on the implicit function theorem to calculate the hypergradient. Solutions are often approximated through conjugate gradient or Neumann series methods. Given a $K$-step approximation, the time complexity is $\mathcal{O}((Km+T)c(d,m))$, as each step involves $m$ matrix-vector products and the whole process is followed by a $T$-step gradient descent The space complexity is $\mathcal{O}(d+m)$.

In practical scenarios, the implicit function update and explicit gradient update methods often yield more precise solutions compared to the explicit proxy update. The latter, while less accurate, is often utilized for its efficiency advantage. As established by Lorraine et al. (2020), the solutions obtained by the implicit function update can be viewed as the limit of the explicit gradient update as the number of iterations $T$ approaches infinity. This insight bridges these two methods. Moreover, Grazzi et al. (2020) demonstrated that both the implicit function update and explicit gradient update methods converge linearly under certain conditions, with the former typically converging at a faster rate. As detailed in Table 6, the time complexity of the implicit function update is $\mathcal{O}((Km+T)c(d,m))$. Compared to the explicit gradient forward's time complexity of $\mathcal{O}(Tmc(d,m))$, the implicit function update method can be more time-efficient when the approximation step $K$ is smaller than $T$ and the hyperparameter dimension $m$ is large. It's worth noting that Lorraine et al. (2020) demonstrate the application of the implicit function update method for optimizing millions of hyperparameters, proving the efficiency of implicit function update. When considering the trade-off between time and memory efficiency, as illustrated in Table 6, the implicit function update is generally more memory-efficient than the reverse mode of the explicit gradient update but somewhat less time-efficient. This analysis serves to guide the choice of method by considering specific requirements of the task at hand, balancing both time and space efficiencies.

Compared to the implicit function update method, the explicit gradient update method is more prevalently employed within the research community, particularly for optimizing hyperparameters such as learning sample weights (Ren et al., 2018; Hu et al., 2019; Wang et al., 2020; Shu et al., 2019). This preference is especially pronounced within the deep learning community, which often leans towards the explicit gradient update method. Several reasons can explain this inclination: (1). The popularity of the explicit gradient update method was propelled by seminal works that used influence functions to study the impact of a training point on prediction outcomes (Koh & Liang, 2017). This method was then widely adopted in subsequent research, particularly in works focusing on learning sample weights (Ren et al., 2018; Hu et al., 2019; Wang et al., 2020; Shu et al., 2019). (2). The explicit gradient update method's appeal lies in its intuitiveness and straightforwardness within the context of deep learning. It can also offer interpretability in certain scenarios. For instance, during sample reweighting, samples with gradients similar to the gradient on the validation set are reweighted to be high (Shu et al., 2019). In contrast, the implicit function update method, while potent, is more complex, relying on optimization techniques such as the conjugate gradient method and Neumann series for approximation. (3). Lastly, the explicit gradient update method benefits from advanced deep learning libraries like higher (Grefenstette et al., 2019), designed explicitly for calculating the explicit gradient update. This tool offers substantial support, making the explicit gradient update method more accessible and efficient to implement, contributing to its prevalence.

**Closed-form update.** The closed-form update method is specifically designed for certain problems where a closed-form solution for the inner level problem can be obtained. Contrary to the other methods, this one yields an exact hypergradient. The most time-consuming aspect of this method is the inverse matrix computation, which results in a time complexity of $\mathcal{O}(n^3)$, where $n$ denotes the dataset size. The space complexity stands at $\mathcal{O}(n^2)$.

Distinguishing between optimizing data and other hyperparameters, we observe divergent preferences in the community regarding the suitable method for each task. For hyperparameters like regularization (Franceschi et al., 2018), the explicit gradient update method is the common choice. This method provides an optimal balance between computational efficiency and solution precision, a crucial consideration for online model training where real-time updates are essential. Its strength lies in its ability to iteratively and efficiently fine-tune hyperparameters, thereby significantly enhancing the model's performance during the training phase. Conversely, the optimization of data, such as data distillation (Nguyen et al., 2020), generally employs the closed-form kernel method. This method may be computationally intensive, leading to perceived inefficiencies, but it delivers highly precise solutions, a quality that is integral in the context of data optimization. In scenarios like data distillation, the output is often used for offline tasks such as continual learning. The computation burden of the closed-form method, therefore, becomes less of a concern because the optimization is carried out once and the solution can be reused indefinitely. This singular computation for lasting usage offsets the computational inefficiency, making it a viable choice for data optimization tasks.

## 5 Conclusion and Future Directions

Bi-level optimization embeds one problem within another and the gradient-based category solves the outer level task via gradient descent methods. We first discuss how to formulate a research problem from a bi-level optimization perspective. There are two formulations: the single-task formulation to optimize hyperparameters and the multi-task formulation to extract meta-knowledge. Further, we discuss four possible ways to compute the hypergradient in the outer level, including explicit gradient update, proxy update, implicit function update, and closed-form update. This could serve as a good guide for researchers on applying gradient-based bi-level optimization.

As we conclude our survey, we spotlight two promising future directions: (1) *Effecctive Data Optimization for Science* from a task formulation perspective corresponding to Section 3. (2) *Accurate Explicit Proxy Update* from an optimization perspective corresponding to Section 4.

### 5.1 Future Direction 1: Effecctive Data Optimization for Science

Depending on the nature of the data, this domain can be primarily divided into three areas: tuning parameter optimization, design optimization, and feature optimization. Here, we elaborate on how bi-level optimization acts as a potent tool to optimize these categories of data, aiding in solving scientific problems.

**Tuning parameter optimization.** In the scientific domain, a common problem is to find a design - be it a protein, a material, a robot, or a DNA sequence - that optimizes a specific black-box objective function such as a protein property score (Trabucco et al., 2022). This process often trains a proxy model using collected pairs of designs and scores along with certain tuning parameters, and the trained proxy guides the design search in evolutionary algorithms (Hansen, 2006) or reinforcement learning (Angermueller et al., 2019). Parameter tuning via bi-level optimization has emerged as a promising approach to achieve data-specific optimization and can enhance the local accuracy of the proxy model around the current point in the search space. This could potentially revolutionize the design search process, making it more effective and targeted. Such optimization can be achieved by locally sampling data points from the collected design-score pairs or by using pseudo-labelers to label neighboring samples. As mentioned in Section 3.1.1, we envision utilizing a clean validation set to fine-tune these data-specific parameters via bi-level optimization. This approach offers promising prospects in identifying reliable portions of locally sampled data, thereby improving the local accuracy of the proxy model, and providing a more effective direction to the search process. As we move forward, the nuances of these techniques will be refined and expanded, offering greater possibilities in design optimization tasks.

**Design optimization.** A paradigm shift in approaching scientific problems could be to focus on directly optimizing the design, instead of tuning parameters to enhance proxy model performance. Data distillation through bi-level optimization, could serve as a pivotal strategy in this context. This technique extracts extensive knowledge from large datasets into distilled samples, a process which has been found to retain features corresponding to the labels associated with the distilled samples (Wang et al., 2018; Lei & Tao,

2023). Taking inspiration from these findings, certain studies assign a predefined high score to an initial design and distill knowledge from the training set into the design (Chen et al., 2022a; 2023a). This procedure results in a design with high-scoring features, essentially creating a desirable final candidate. As we look towards the future of gradient-based bi-level optimization, two intriguing directions arise from this research line. Firstly, employing foundational models specific to fields such as materials science, alongside advanced data distillation techniques (Lei & Tao, 2023), could substantially improve distillation performance. This would enable more accurate high-scoring features to be incorporated into the design. Secondly, analyzing the distilled high-scoring sample to extract patterns and potential rules presents another exciting prospect. Armed with this derived knowledge, it may be possible to create high-scoring designs or to integrate this understanding into our modelling process, paving the way for more insightful and accurate predictions.

**Feature optimization.** Another pivotal area in science is feature optimization, which aims to improve the learning of data features. This optimization process is intertwined with scientific constraints, which can be effectively incorporated as the inner level task in bi-level optimization. These constraints generally include geometric and biochemical constraints. Geometric constraints stem from the physical or spatial characteristics of a system. An example of leveraging these constraints can be seen in the work (Xu et al., 2021). Here, the process of molecular conformation prediction is decomposed into two levels, with the geometric constraint between molecular conformation and predicted atom distances being the inner level task. By utilizing this constraint, the trained neural network can produce atom distance features that are more compatible with the actual molecular structure. Biochemical constraints are rooted in the chemical and biological principles that govern a system. These constraints have been utilized by Chen et al. (2022b), where the inherent correspondence between the sequential and structural representations of a protein is used as the biochemical constraint. The task of maximizing mutual information is formulated as the inner level task, which allows the Graph Neural Network (GNN) to output protein structural features with higher accuracy. Moving forward, feature optimization is set to play an increasingly important role in enhancing the learning of data features by leveraging inherent scientific constraints within a bi-level optimization framework.

## 5.2   Future Direction 2: Accurate Explicit Proxy Update

Beyond the potential identified in the first future direction, another significant area of exploration centers around constructing a more accurate proxy in explicit proxy update. This promises to yield a more accurate computation of hypergradients efficiently. One viable strategy is to employ a more interpretable construction of the proxy network, $\boldsymbol{\theta}^* = P_\alpha(\boldsymbol{\phi})$. For example, Bohdal et al. (2021) design a proxy by utilizing a weighted average of the perturbed inner variable, a process that offers interpretability and improved accuracy.

Moreover, we underline the connection between model-based optimization (Trabucco et al., 2022) and explicit proxy update, in the context of computing the inner variable from the outer variable using gradient methods. Specifically, the explicit proxy update aims to discover a proxy network that maps the outer variable to the inner variable. This mapping bears a resemblance to the trained model in model-based optimization, which maps the input design (e.g., a robot) to its property (e.g., robot speed). Consequently, recent advancements in model-based optimization can potentially enhance the performance of explicit proxy update. Here, we discuss some directions that are not exhaustive:

**Modeling Priors.** In model-based optimization, integrating various modeling priors, such as smoothness (Yu et al., 2021), is critical for model performance. Analogously, these priors could be integrated into the training of the proxy network, leading to more precise prediction from the outer to the inner variable.

**Importance Sampling.** Model-based optimization utilizes data gathered during the optimization process, and the model might not be accurate for the neighborhood of the current optimization point. A technique used in model-based optimization (Fannjiang & Listgarten, 2020) involves employing importance sampling to retrain the model, based on input distribution. This approach could be adapted to the training of proxy neural network on the already gathered outer-inner variable pairs.

**Reverse Mapping.** Some works in model-based optimization (Fannjiang & Listgarten, 2020; Chan et al., 2021) suggest using a reverse mapping for predicting the input design from the property score using generative modeling techniques. By maintaining consistency between the forward mapping (i.e., the model prediction process) and the reverse mapping, the model's accuracy is significantly enhanced. A parallel strategy can be

applied in explicit proxy updates where the introduction of a reverse mapping to predict the outer variable from the inner variable could improve the accuracy of the proxy by ensuring consistency.

**Efficient Sampling.** Training the proxy neural network involves sampling the outer variable and computing the corresponding inner variable, which can be computationally expensive. Utilizing the acquisition function, as seen in model-based optimization (Trabucco et al., 2022), to decide the next batch for sampling may prove to be an efficient solution to this challenge.

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
