# OpenReview forum: "Gradient-based Bi-level Optimization for Deep Learning: A Survey"
_TMLR — Rejected by TMLR_

### Review · Reviewer_wUmb · 2023-05-22

**Summary Of Contributions:**

This review covers a range of recent applications of bi-level optimization in machine learning problems. The paper consists of two principal parts: the first illustrates a variety of means by which a bi-level optimization problem can be constructed in several application areas, ranging from game theory to computational biology; the second provides examples of how the hypergradient of a bi-level optimization problem can be computed. The principal contribution of the paper is in consolidating these varied methodologies into a single work, providing a high-level overview of how bi-level optimization can be instantiated and implemented.

**Audience:**

Yes

**Broader Impact Concerns:**

No braoder impact concerns

**Claims And Evidence:**

Yes

**Requested Changes:**

Requested changes
  - In Section 3, the notation used to describe each concrete instantiation of the bi-level optimization problem is quite repetitive and doesn't always do a good job at illustrating the specifics of the problem by e.g. formalizing inline what the set being optimized over is or what form the objective function takes. As a result, we often see a paragraph of text describing an optimization problem, followed by the same \max \min R^{\out} as is presented in the first presentation of bi-level optimization problems.
  - The discussion of hypergradient computation with Lagrange multipliers is missing several details that are necessary to understand how the final form of th eobjective is constructed. Since the goal of the paper is to arm practitioners with an understanding of how to apply bi-level optimization methods to their own problem settings, this seems like an important gap to fill.
  -  The discussion of the closed-form solution for ridge regression, in contrast, could easily be compressed if space is a concern as the closed form of the least-squares solution is well known.
  - It isn't clear to me how the Stackelberg game relates to *gradient* optimization

Questions:
  - How practical are the continuous-time models and ODE solutions in practical settings?
  - Some methods use several inner-loop optimizatoin steps, whereas others only use a single inner loop step to approximate the inner optimization. How does the suboptimality of the inner loop output influence the accuracy and performance of these methods?

**Strengths And Weaknesses:**

**Strengths:**
- The figures in Sections 3 and 4 are clear and help to convey the gist of how the illustrated methods work.
- The paper conveys many examples which cover a wide breadth of the existing approaches
- The partitioning of the paper into the formulation of a bi-level optimization problem and the consequent computation of hypergradients made sense and made for a clear story.

**Weaknesses:**
  - The paper illustrates how many existing methods can be viewed as bi-level optimization, but it's not always clearly explained why the particular formulation is desirable or whether, for example, recent developments have improved the computational complexity or reliability of the method.
  - In general, I would have liked to see more direct comparisons between methods of computing hypergradients in Section 4, and more of a discussion on when one class of methods might be preferrable over the other. The usefulness of the current version of the paper is significantly limited by this lack of comparison, as the reader has little information to go by on whether a particular hypergradient computation approach will be useful in their problem setting.
  - The writing of the paper has a number of noticeable issues, particularly the typesetting of citations: several citations are missing a space between the prior word and the start of the citation, and the use of \citet where \citep would have been more appropriate is frequent.

---

> ### Author Response · Authors · 2023-06-10
> **Response to review questions 1/n**
>
> ## Weaknesses
>
> > The paper illustrates how many existing methods can be viewed as bi-level optimization, but it's not always clearly explained why the particular formulation is desirable or whether, for example, recent developments have improved the computational complexity or reliability of the method.
>
> We greatly appreciate your constructive feedback. We acknowledge the necessity of explaining the appeal of certain bi-level optimization formulations and the advancements recent developments bring, particularly in terms of computational complexity.
>
> In response to your comments, we have meticulously revised Section 3 to accentuate the reasons for favoring specific formulations. We have also introduced a new section, Section 4.5, that provides a detailed discussion on computational complexity.
>
> We believe that these modifications will provide a more comprehensive understanding of the discussed works, their unique contributions, and broader implications. We hope these revisions address your concerns effectively, and we welcome further suggestions to enhance the clarity and value of our paper.
>
> > In general, I would have liked to see more direct comparisons between methods of computing hypergradients in Section 4, and more of a discussion on when one class of methods might be preferrable over the other. The usefulness of the current version of the paper is significantly limited by this lack of comparison, as the reader has little information to go by on whether a particular hypergradient computation approach will be useful in their problem setting.
>
> We appreciate your valuable feedback. To address your concern about the need for a direct comparison of the hypergradient computation methods, we have incorporated a new subsection, Section 4.5, into the paper. This section is dedicated to offering a comparative analysis of these methods, primarily focusing on aspects such as time complexity, space complexity, and solution accuracy. Furthermore, in this new section, we delve into discussing situations where one class of methods may be preferable over the other.  Due to the extensive use of mathematical symbols in this section, it is challenging to accurately display the content in the markdown format. Hence, we would kindly direct you to Section 4.5 in the updated manuscript for a comprehensive comparison and analysis. We believe this new addition will address your concern and greatly enhance the usefulness of our paper.
>
> > The writing of the paper has a number of noticeable issues, particularly the typesetting of citations: several citations are missing a space between the prior word and the start of the citation, and the use of \citet where \citep would have been more appropriate is frequent.
>
> We appreciate your careful observation regarding the citation styles and the occasional missing spaces in the manuscript. In response to your feedback, we have meticulously reviewed the entire document and made necessary revisions. We have corrected all instances where spaces were missing before the citations, and have ensured the proper usage of the \citet and \citep commands as per the citation context. We are confident that these modifications have enhanced the overall readability and professionalism of the paper. Thank you for bringing these issues to our attention.
>
> ## Requested Changes
>
> > In Section 3, the notation used to describe each concrete instantiation of the bi-level optimization problem is quite repetitive and doesn't always do a good job at illustrating the specifics of the problem by e.g. formalizing inline what the set being optimized over is or what form the objective function takes. As a result, we often see a paragraph of text describing an optimization problem, followed by the same \max \min R^{\out} as is presented in the first presentation of bi-level optimization problems.
>
>
> We appreciate your insightful comments about the repetitive notation and lack of specificity in illustrating each bi-level optimization problem in Section 3. Taking your feedback into account, we have thoroughly revised this section.
>
> We've strived to ensure each unique instantiation of the bi-level optimization problem is now detailed more precisely, clearly delineating the objective function and the set being optimized over. We've also made a concerted effort to streamline the use of notation, thereby eliminating redundancy. These changes have been reinforced by incorporating several tables for a more organized presentation.
>
> With these modifications, we trust the clarity and readability of this section have been significantly enhanced, thus offering a more comprehensive understanding of each bi-level optimization problem.

---

> > ### Author Response · Authors · 2023-06-10
> > **Response to review questions 2/n**
> >
> > > The discussion of hypergradient computation with Lagrange multipliers is missing several details that are necessary to understand how the final form of th eobjective is constructed. Since the goal of the paper is to arm practitioners with an understanding of how to apply bi-level optimization methods to their own problem settings, this seems like an important gap to fill.
> >
> >
> > Thank you for pointing out the lack of detail in the section discussing hypergradient computation using Lagrange multipliers. We understand your concern and agree that this is crucial for our audience, who are likely interested in applying bi-level optimization methods to their own problem settings. To address this, we have augmented this section with more comprehensive details to clarify how the final form of the objective is constructed. We believe these additions will provide readers with a more solid understanding of the procedure and how they can implement it in their own work.
> >
> > > The discussion of the closed-form solution for ridge regression, in contrast, could easily be compressed if space is a concern as the closed form of the least-squares solution is well known.
> >
> >
> > Thank you for your suggestion on the section discussing the closed-form solution for ridge regression. We agree with you that the closed form of the least-squares solution is widely known in the field. Hence, to maintain focus and utilize space more efficiently, we have streamlined this section accordingly, ensuring the essential details remain clear and comprehensible.
> >
> > > It isn't clear to me how the Stackelberg game relates to gradient optimization
> >
> > Thank you for your feedback and the opportunity to clarify the connection between the Stackelberg game and gradient-based bi-level optimization. Indeed, the Stackelberg game can be viewed as an instance of a bi-level optimization problem. Specifically, the quantity of products produced by the leader company ($\phi$) can be seen as the "outer" variable, and the quantity produced by the follower company ($\theta$) is the "inner" variable.
> >
> > In the context of gradient-based bi-level optimization, the connection between these two variables is established by the inner level problem, and this differentiable connection is used to compute the hypergradient of the outer objective function with respect to the outer variable $\phi$. This gradient is then used to update the outer variable $\phi$.
> >
> > In other words, the Stackelberg game demonstrates the concept of gradient-based bi-level optimization by showcasing how the leader company optimizes its strategy (the outer level problem) while taking into account the follower company's reaction (the inner level problem), and how the leader's strategy is adjusted based on the gradients computed from this interrelationship.
> >
> > We acknowledge that the original manuscript may not have made this connection explicitly clear. Therefore, in the revised version of the paper, we have added additional explanatory text to highlight this relationship and clarify the link between the Stackelberg game and gradient-based bi-level optimization. We hope this enhanced explanation provides a clearer understanding for readers.

---

> > > ### Author Response · Authors · 2023-06-10
> > > **Response to review questions 3/n**
> > >
> > > ## Questions
> > >
> > > > How practical are the continuous-time models and ODE solutions in practical settings?
> > >
> > > The application of continuous-time models and ODE solutions in practical settings has been demonstrated to be quite efficient and competitive in terms of both computational time and memory usage. As shown in the study [1], when tested on a single 5-shot 5-way task using a Conv-4 backbone, the continuous-time model exhibited superior efficiency compared to both MAML and iMAML. Furthermore, despite this efficiency, the model still managed to achieve a level of accuracy on par with these other methods.
> > >
> > >
> > >
> > > > Some methods use several inner-loop optimizatoin steps, whereas others only use a single inner loop step to approximate the inner optimization. How does the suboptimality of the inner loop output influence the accuracy and performance of these methods?
> > >
> > > The degree of suboptimality in the inner loop output can significantly impact the overall accuracy and performance of bi-level optimization methods. According to the study [2], where experiments were conducted on 100-way classification tasks to optimize feature map hyperparameters, it was observed that the inner level solution's accuracy generally improved with the increase in inner-loop optimization steps (T=1, 4, 16, 64, 256). Furthermore, the performance of the outer-level hyperparameter optimization also demonstrated a positive correlation with the accuracy of the inner level solution, reaching around 100\% after only T=4 steps. However, it is worth noting that the increase in accuracy comes at the expense of higher computational time on NVidia Tesla M40 GPU, with times for T=1, 4, 16, 64, 256 being 60sec, 119sec, 356sec, 1344sec, and 5532sec, respectively. Thus, while increasing the number of inner-loop steps can enhance performance and accuracy, it does also increase the computational requirements.
> > >
> > > ## In Summary
> > > Does our response adequately address your concerns? We want to express our sincere gratitude for your comprehensive review and invaluable feedback. Your careful scrutiny of our paper has been essential for its improvement. We look forward to more constructive exchanges in the upcoming rebuttal phase.
> > >
> > >
> > >     [1] Tristan Deleu, David Kanaa, Leo Feng, Giancarlo Kerg, Yoshua Bengio, Guillaume Lajoie, and Pierre-Luc Bacon. Continuous-time meta-learning with forward mode differentiation. In International Conference on Learning Representations, 2022. URL https://openreview.net/forum?id=57PipS27Km.
> > >
> > >     [2] Luca Franceschi, Paolo Frasconi, Saverio Salzo, Riccardo Grazzi, and Massimiliano Pontil. Bilevel programming for hyperparameter optimization and meta-learning. In International Conference on Machine Learning, 2018.

---

> > > > ### Author Response · Authors · 2023-06-17
> > > > **Looking forward to your feedback**
> > > >
> > > > Dear Reviewer,
> > > >
> > > > I trust this message finds you well. In line with your valuable feedback, we have made several enhancements to our manuscript. I am outlining these changes below to ensure that all your concerns have been appropriately addressed.
> > > >
> > > > **Weaknesses**
> > > >
> > > > 1. **Explanation of Bi-level Formulations:** In response to your suggestion, we have enriched Section 3 to clarify the appeal of specific bi-level optimization formulations. Additionally, we have introduced a new section (Section 4.5) that discusses computational complexity in depth.
> > > >
> > > > 2. **Comparison between Methods of Computing Hypergradients:** Following your recommendation, we have incorporated a new subsection (Section 4.5) that offers a comprehensive comparison between different hypergradient computation methods. It also discusses the situations in which one class of methods might be preferred over the other.
> > > >
> > > > 3. **Citation Typesetting Issues:** Upon your observation, we have revised the entire manuscript to correct citation typesetting issues. This includes ensuring proper usage of \citet and \citep commands and fixing missing spaces before citations.
> > > >
> > > > **Requested Changes**
> > > >
> > > > 1. **Repetition in Notation and Specificity of Bi-level Optimization Problem Descriptions in Section 3:** As you advised, we have amended Section 3 to eliminate redundancy in notation and to enhance the specificity of each bi-level optimization problem description. We have also added tables for better organization.
> > > >
> > > > 2. **Missing Details in Hypergradient Computation with Lagrange Multipliers:** Based on your feedback, we have expanded the discussion about hypergradient computation using Lagrange multipliers to provide a clearer picture of how the final form of the objective is constructed.
> > > >
> > > > 3. **Compression of Ridge Regression Closed-form Solution:** Following your suggestion, we have condensed the discussion of the closed-form solution for ridge regression, ensuring the necessary details remain clear and comprehensible.
> > > >
> > > > 4. **Relating Stackelberg Game to Gradient Optimization:** In response to your query, we have enhanced the explanation of how the Stackelberg game relates to gradient-based bi-level optimization. The revised manuscript now provides a clearer connection between these two concepts.
> > > >
> > > > **Answers to Questions**
> > > >
> > > > 1. **Practicality of Continuous-time Models and ODE Solutions:** In answer to your question, we have highlighted how continuous-time models and ODE solutions have proven to be efficient and competitive in practical scenarios.
> > > >
> > > > 2. **Impact of Suboptimality of Inner Loop Output on Accuracy and Performance:** To address your question, we have discussed how the degree of suboptimality in the inner loop output significantly influences the overall accuracy and performance of bi-level optimization methods.
> > > >
> > > > We hope that these updates adequately address your feedback. Please do let us know if you believe further adjustments are necessary.
> > > >
> > > > Best Regards,
> > > >
> > > > Paper 1065 Authors.

---

> > > > > ### Author Response · Authors · 2023-06-24
> > > > > **Friendly Reminder: Feedback on Revised Submission**
> > > > >
> > > > > Dear Reviewer,
> > > > >
> > > > > I hope this message finds you well.
> > > > >
> > > > > I wanted to take a moment to remind you that the deadline for feedback on our revised manuscript is approaching. It's slated for two days from today. Your initial review was insightful and helped us to improve the quality of our paper significantly. We have addressed your comments in our revision and are looking forward to receiving your feedback on these changes.
> > > > >
> > > > > We understand that you are busy, and we greatly appreciate the time and effort you put into reviewing our work. Thank you once again for your support and contribution to improving our work.
> > > > >
> > > > > Best regards,
> > > > >
> > > > > Paper 1065 Authors.

---

> > > > > > ### Comment · Reviewer_wUmb · 2023-06-27
> > > > > > **Response to Updates**
> > > > > >
> > > > > > Thanks to the authors for uploading a comprehensive set of revisions to the paper, and for their patience as it took me some time to thoroughly review the substantial edits. I appreciate the addition of several tables summarizing results and comparing and contrasting different methods. I agree with several of the other reviewers' points concerning e.g. referring to explicit outer-level hypergradients in the taxonomy before they are formally discussed. While the additional discussion and comparisons do improve the paper, the overall message is still a bit unclear. In general, a survey paper covers a particular topic or methodology in a way that allows newcomers to the field to quickly get an overview of a set of related methods and understand their relative strengths and weaknesses along with their use-cases. Bi-level optimization to me presents a challenging topic to write a good survey paper on, as the specific problems it can be applied to spread across a broad range of disconnected research communities, and the methodologies used for different problem formulations are quite distinct. To write a compelling survey paper on such a diverse and disconnected literature, it is critical to convey to the reader _why_ they should be interested in understanding how all of these different problem settings and methods fit together. In its current form I think the paper still doesn't do a great job at conveying this, and still comes across a bit as a potpourri of disparate problems and tools which all share the loose connection of being expressed as an optimization over an inner and an outer loop, without a clear sense of what is gained by viewing them in aggregate. For example, is it the case that mathematically similar bi-level optimization problems in different subfields tend to be solved using different methods by sheer accident of history wherein a research community overfits to one set of methods that happened to be published first within their subfield? Could tools from one discipline potentially be applied to another? This type of insight would significantly increase the value of the survey's contribution and is still somewhat lacking at the moment.

---

> > > > > > > ### Comment · Reviewer_UciL · 2023-06-28
> > > > > > > **Very good summary of paper!**
> > > > > > >
> > > > > > > Very well written Reviewer wUmb! I think you summarized my feelings about the paper better than I did in my own response 🥲
> > > > > > >
> > > > > > > To re-phrase what you wrote in another way: maybe it would help if the paper presented a more convincing case for _why_ we should consider these disparate problems in aggregate, beyond just noting that the mathematical form is similar. I do not think the paper clearly answers this in its current form.

---

> > > > > > > > ### Author Response · Authors · 2023-06-28
> > > > > > > > **Response to review questions 1/n**
> > > > > > > >
> > > > > > > > > Thanks to the authors for uploading a comprehensive set of revisions to the paper, and for their patience as it took me some time to thoroughly review the substantial edits. I appreciate the addition of several tables summarizing results and comparing and contrasting different methods. I agree with several of the other reviewers' points concerning e.g. referring to explicit outer-level hypergradients in the taxonomy before they are formally discussed. While the additional discussion and comparisons do improve the paper, the overall message is still a bit unclear. In general, a survey paper covers a particular topic or methodology in a way that allows newcomers to the field to quickly get an overview of a set of related methods and understand their relative strengths and weaknesses along with their use-cases. Bi-level optimization to me presents a challenging topic to write a good survey paper on, as the specific problems it can be applied to spread across a broad range of disconnected research communities, and the methodologies used for different problem formulations are quite distinct. To write a compelling survey paper on such a diverse and disconnected literature, it is critical to convey to the reader why they should be interested in understanding how all of these different problem settings and methods fit together. In its current form I think the paper still doesn't do a great job at conveying this, and still comes across a bit as a potpourri of disparate problems and tools which all share the loose connection of being expressed as an optimization over an inner and an outer loop, without a clear sense of what is gained by viewing them in aggregate. For example, is it the case that mathematically similar bi-level optimization problems in different subfields tend to be solved using different methods by sheer accident of history wherein a research community overfits to one set of methods that happened to be published first within their subfield? Could tools from one discipline potentially be applied to another? This type of insight would significantly increase the value of the survey's contribution and is still somewhat lacking at the moment.
> > > > > > > > > Very well written Reviewer wUmb! I think you summarized my feelings about the paper better than I did in my own response. To re-phrase what you wrote in another way: maybe it would help if the paper presented a more convincing case for why we should consider these disparate problems in aggregate, beyond just noting that the mathematical form is similar. I do not think the paper clearly answers this in its current form.
> > > > > > > >
> > > > > > > > We appreciate the comprehensive feedback and the time you took to evaluate the revisions. While acknowledging the valid concerns raised about the overall structure and goal of the paper, we aim to address these points as follows.
> > > > > > > >
> > > > > > > > 1. We've expanded Section 4.5 to better elucidate why the explicit gradient update method is widely utilized in hyperparameter optimization, particularly in the deep learning community.
> > > > > > > >
> > > > > > > > "Compared to the implicit function update method, the explicit gradient update method is more prevalently employed within the research community, particularly for optimizing hyperparameters such as learning sample weights. This preference is especially pronounced within the deep learning community, which often leans towards the explicit gradient update method. Several reasons can explain this inclination: (1). The popularity of the explicit gradient update method was propelled by seminal works that used influence functions to study the impact of a training point on prediction outcomes. This method was then widely adopted in subsequent research, particularly in works focusing on learning sample weights. (2). The explicit gradient update method's appeal lies in its intuitiveness and straightforwardness within the context of deep learning. It can also offer interpretability in certain scenarios. For instance, during sample reweighting, samples with gradients similar to the gradient on the validation set are reweighted to be high. In contrast, the implicit function update method, while potent, is more complex, relying on optimization techniques such as the conjugate gradient method and Neumann series for approximation. (3). Lastly, the explicit gradient update method benefits from advanced deep learning libraries like higher, designed explicitly for calculating the explicit gradient update. This tool offers substantial support, making the explicit gradient update method more accessible and efficient to implement, contributing to its prevalence."

---

> > > > > > > > > ### Author Response · Authors · 2023-06-28
> > > > > > > > > **Response to review questions 2/n**
> > > > > > > > >
> > > > > > > > > 2. We've observed that different subfields like data optimization and hyperparameter optimization have distinct solver preferences. We've detailed these insights in Section 4.5:
> > > > > > > > >
> > > > > > > > > "Distinguishing between optimizing data and other hyperparameters, we observe divergent preferences in the community regarding the suitable method for each task. For hyperparameters like regularization, the explicit gradient update method is the common choice. This method provides an optimal balance between computational efficiency and solution precision, a crucial consideration for online model training where real-time updates are essential. Its strength lies in its ability to iteratively and efficiently fine-tune hyperparameters, thereby significantly enhancing the model's performance during the training phase. Conversely, the optimization of data, such as data distillation, generally employs the closed-form kernel method. This method may be computationally intensive, leading to perceived inefficiencies, but it delivers highly precise solutions, a quality that is integral in the context of data optimization. In scenarios like data distillation, the output is often used for offline tasks such as continual learning. The computation burden of the closed-form method, therefore, becomes less of a concern because the optimization is carried out once and the solution can be reused indefinitely. This singular computation for lasting usage offsets the computational inefficiency, making it a viable choice for data optimization tasks."
> > > > > > > > >
> > > > > > > > > 3. We've further explained our reasoning behind the criteria used in single-task classification in Section 3.1:
> > > > > > > > >
> > > > > > > > > "(1) A notable situation is when the constraint and the main optimization problem use different mathematical formulas, implying that they optimize entirely different problems. In these scenarios, the constraint typically arises organically, and its identification becomes straightforward. Examples include the energy constraint present in topology design or the bio-chemical constraint in protein representation learning. Conversely, when the main optimization problem and the constraint share the same mathematical formula, the formula needs to be broken down into two levels. This breakdown is usually accomplished by considering data variations between training and validation sets. The inner level, typically represented by the training loss, functions as the constraint in this setting. This comprises the first criterion for our evaluation. (2) In some cases, the main optimization problem might not directly contain the outer variable. In such cases, the connection built at the inner level is utilized. This situation poses a challenge in formulating the outer level task, thereby leading us to introduce a second criterion. This second criterion classifies works based on whether the calculation of the hypergradient relies exclusively on the established inner level connection."
> > > > > > > > >
> > > > > > > > > We hope these revisions and explanations clarify the paper's aim to present a comprehensive survey on bi-level optimization, connecting disparate problem settings and methods under a shared framework.

---

### Review · Reviewer_UciL · 2023-05-23

**Summary Of Contributions:**

This is a review paper which reviews surveys methods for bi-level optimization in deep learning and its applications to practical problems. As such, it doesn't really make explicit claims. Quoting the introduction, I think this paper's intended contribution is:

> This survey aims to guide researchers on their research problems involving bi-level optimization.

**Audience:**

No

**Broader Impact Concerns:**

no concerns for broader impact

**Claims And Evidence:**

No

**Requested Changes:**

I would definitely like the clarity of the paper to be improved, but even if that were the case, I am actually not sure what it would take to secure my recommendation of acceptance to TMLR. Reading the [evaluation criteria](https://jmlr.org/tmlr/editorial-policies.html#evaluation), I noticed that there is no place for survey papers (even very clear ones): TMLR seems to want each paper to contain some form of new knowledge. I think the criterion that could best apply to this paper is:

> accounts of applications of existing techniques that shed light on the strengths and weaknesses of the methods;

However, as I mentioned above I don't think that this paper does this. The authors themselves sought to "guide researchers on their research problems involving bi-level optimization", but I don't think this paper really does so, and I don't see an easy fix for this without significantly changing the focus of the paper. So my overall requested change would be "do something which adds new knowledge to the field or explain what new knowledge your paper does contribute which I missed during my review". Some ideas for this are:

- compare/contrast various approximate bilevel optimization methods in terms of speed, accuracy, and memory
- discuss pros/cons of implicit vs explicit bilevel optimization methods
- discuss for which problems it is most suitable to formalize them as bilevel optimization methods. For example, hyperparmeter optimization could be formalized in other ways besides bilevel optimization. It might not be the best tool for every problem.

**Strengths And Weaknesses:**

I think the biggest strength of this paper is its coverage of a lot of work using bi-level optimization. I have done work in this area but was not aware of 50%+ of the papers that they described. However, I think the key weaknesses of this paper are its general lack of clarity and the lack of insights or new knowledge. I will expand on these two points below.

### Lack of clarity

I thought the paper was confusing to read. It is hard to attribute this to any one particular part of the paper, but I will give some examples below:

- Unclear in equations 1-2 why the outer loop should always use the validation set and the inner loop should always use the training set. I know many papers do this but it doesn't seem fundamental to bilevel optimization.
- "meta knowledge" is an unclear term. I did not know what it means, and searching on Google scholar I found only a few search results for it which mostly referred to "knowledge about knowledge". Did the authors intend to write "meta learning" instead? It also wasn't clear to me whether bilevel optimization for meta-learning is really that different to bilevel optimization for other applications.
- Von Stackelberg games were mentioned a lot in the introduction but not explained until section 3.1.2. I think most readers in ML will not be familiar with this idea (since it seems to come from economics) so I think its best to not mention it until it is explained properly.
- The categorization of methods with and without "explicit hypergradients" was unclear. What is an "explicit hypergradient?" Is this whether $\frac{\partial L_{out}}{\partial \phi}=0$? If so then it would probably make sense to move section 4 before section 3 so that this is introduced.
- In general, most of the methods surveyed were described very briefly and for many of them it was unclear exactly how the authors used bilevel optimization or how their formalism related to other methods.

### Lack of insights or new knowledge

The paper seemed to mostly be a list of other works that have used bilevel optimization without additional analysis or insights relating these works to each other or pointing out common challenges. Reading section 3 felt like "X et al used bilevel optimization for Y, A et al used it for Z, etc". I thought that this could more or less go into a table. In fact, that might even be more readable than listing it out explicitly. I did not feel like there were any unifying insights or insights about commonalities between these works.

At the same time, although section 4 surveyed some methods to perform bilevel optimization, I felt like it completely missed what I view as the key challenge in this area: the trade-off between memory, computation time, and accuracy. Computing exact gradient updates tends to be expensive in either time or memory, and approximations reduce this at the cost of error. For large systems approximations are generally necessary, but it is not clear whether these approximations prevent good solutions from being found. A lot of works sort of dance around this issue so it would be nice for it to be tackled more directly in a review paper.

---

> ### Author Response · Authors · 2023-06-10
> **Response to review questions 1/n**
>
> ## General Reply
> We greatly appreciate your insightful and constructive feedback that has guided us in improving the quality of our paper. Your detailed comments have assisted us in identifying areas that needed clarification, correction, and enhancement. Following your suggestions, we have meticulously revisited and revised the paper, as detailed in the subsequent sections. Your time and effort in reviewing our work is sincerely valued.
> ## Lack of clarity
> > Unclear in equations 1-2 why the outer loop should always use the validation set and the inner loop should always use the training set. I know many papers do this but it doesn't seem fundamental to bilevel optimization.
>
> Thank you for your insightful comment regarding the use of the validation set in the outer loop and the training set in the inner loop in equations 1-2. We agree with your perspective that this setup isn't necessarily fundamental to bi-level optimization itself. To maintain clarity and avoid potential misunderstanding, we have decided to remove the specific labels "training" and "validation" from the equations.
>
> > "meta knowledge" is an unclear term. I did not know what it means, and searching on Google scholar I found only a few search results for it which mostly referred to "knowledge about knowledge". Did the authors intend to write "meta learning" instead? It also wasn't clear to me whether bilevel optimization for meta-learning is really that different to bilevel optimization for other applications.
>
> We appreciate your query about the term "meta knowledge." In the context of our paper, "meta knowledge" is indeed analogous to the concept of "meta-learning," in that it refers to knowledge that is acquired across various tasks, rather than being confined to a single task. This usage is outlined in Section 2 of our manuscript, where "meta knowledge" is described as information or insights that extend beyond the scope of individual tasks.
>
> We classify bi-level optimization for meta-learning and bi-level optimization for hyperparameter optimization as distinct research areas, unified under the broader umbrella of bi-level optimization. This classification, aligned with the approach taken in [1], allows researchers unfamiliar with one area to get a clear overview of both, highlighting their interconnection. The primary distinction between these two applications of bi-level optimization lies in the nature of the knowledge being pursued. In many applications of bi-level optimization, the objective is to learn task-specific hyperparameters. Conversely, when bi-level optimization is applied to meta-learning, the goal is to acquire more generalized knowledge that is applicable across a variety of tasks. We hope this explanation clarifies our usage of the term "meta knowledge" and the underlying rationale for our classification of bi-level optimization strategies.
>
>
> > Von Stackelberg games were mentioned a lot in the introduction but not explained until section 3.1.2. I think most readers in ML will not be familiar with this idea (since it seems to come from economics) so I think its best to not mention it until it is explained properly.
>
> Thank you for your thoughtful suggestion regarding the presentation of Von Stackelberg games in our paper. We understand that it might be a less familiar concept for many readers in the field of machine learning, as it originates from economics. In line with your feedback, we have carefully revised our manuscript to eliminate most mentions of Stackelberg games prior to Section 3.1.2. This way, the concept will be introduced and explained comprehensively before its applications are discussed. We believe this adjustment will make the manuscript more accessible and straightforward to our readers.
>
> > The categorization of methods with and without "explicit hypergradients" was unclear. What is an "explicit hypergradient?" Is this whether the partial deriviate is zero? If so then it would probably make sense to move section 4 before section 3 so that this is introduced.
>
> Thank you for your observation regarding the classification of methods with and without "explicit hypergradients". An "explicit hypergradient" refers to cases where the outer level loss directly includes a certain type of hyperparameter, which then allows for the direct computation of the hypergradient. As an illustration, the DARTS case entails an explicit hypergradient because the outer level loss directly involves the network architecture. Conversely, in the instance reweighting scenario, the outer level loss does not directly contain the instance weights; thus, the gradient must be calculated implicitly via the inner level connection. While we acknowledge that the examples in Section 4 do not feature explicit hypergradients, we have chosen to introduce the single-task formulation in Section 3 before proceeding to these examples. We believe this sequence allows for a more logical and clear progression through the different aspects of our study.

---

> > ### Author Response · Authors · 2023-06-10
> > **Response to review questions 2/n**
> >
> > > In general, most of the methods surveyed were described very briefly and for many of them it was unclear exactly how the authors used bilevel optimization or how their formalism related to other methods.
> >
> > Thank you for your insightful feedback. We acknowledge the necessity for a more detailed presentation of the surveyed methods and their specific application of bilevel optimization. In response to your comments, we have expanded upon these methods in our revision, clarifying how each one is formulated within the framework of bilevel optimization. We believe these modifications will enhance the clarity of our paper and foster a better understanding of the diverse applications of bilevel optimization.
> >
> > ## Lack of insights or new knowledge
> >
> > > The paper seemed to mostly be a list of other works that have used bilevel optimization without additional analysis or insights relating these works to each other or pointing out common challenges. Reading section 3 felt like "X et al used bilevel optimization for Y, A et al used it for Z, etc". I thought that this could more or less go into a table. In fact, that might even be more readable than listing it out explicitly. I did not feel like there were any unifying insights or insights about commonalities between these works.
> >
> > We appreciate your constructive feedback. Based on your suggestions, we have undertaken a comprehensive revision of Section 3, structuring it with tables for an easier overview of the works involving bilevel optimization. We've aimed to make it more readable rather than being a simple enumeration of various studies. Moreover, we've endeavored to add our unique insights and highlight common threads running through these works. For instance, during our discussion on DARTS, we underscored the significant role played by the conversion of discrete variables into continuous ones in gradient-based bilevel optimization. "It's notable that the conversion of discrete variables to continuous ones holds significant importance in gradient-based bi-level optimization as it offers an efficient method to adjust discrete parameters. This technique finds its application in earlier label learning methodologies, as demonstrated in the works by \citep{algan2021meta, wu2021learning}. In these cases, discrete one-hot labels are morphed into soft labels for optimization. However, in such instances, the conversion to continuous forms isn't as crucial as in our present context, owing to the existence of alternate methods like instance reweighting for managing label noise."
> >
> > Furthermore, in our discussion on instance reweighting, we spotlighted the necessity of a neural network for parameterizing high-dimensional hyperparameters to ensure efficient updates, "To circumvent this issue, subsequent works~\citep{shuMetaWeightNetLearningExplicit2019a} devise an alternative solution - a weighting network. This network is designed to parameterize instance weights, wherein the input is the instance loss, and the output becomes the instance weight, enhancing scalability for large datasets. This innovative approach has been further employed by \citet{xu2021end}, who parameterized the atom distance in a molecular structure using a message passing neural network, a mechanism designed to encapsulate graph information effectively. It is crucial to note that efficient optimization is closely tied to the effective parameterization of high-dimensional hyperparameters. Achieving this involves the use of a meticulously designed neural network/input that caters to the demands of the specific problem at hand, ensuring both efficiency and accuracy in the optimization process."
> >
> > > At the same time, although section 4 surveyed some methods to perform bilevel optimization, I felt like it completely missed what I view as the key challenge in this area: the trade-off between memory, computation time, and accuracy. Computing exact gradient updates tends to be expensive in either time or memory, and approximations reduce this at the cost of error. For large systems approximations are generally necessary, but it is not clear whether these approximations prevent good solutions from being found. A lot of works sort of dance around this issue so it would be nice for it to be tackled more directly in a review paper.
> >
> > Thank you for raising this important point. We fully agree that the trade-off between memory usage, computation time, and accuracy is indeed a crucial challenge in the field of bilevel optimization. In response to your comments, we have added Section 4.5, which specifically focuses on this trade-off, and explores how different methods manage this balance. By doing so, we aim to address this critical issue head-on and provide a more comprehensive and practical view on the subject. We hope that this addition will enhance the depth and utility of the review.

---

> > > ### Author Response · Authors · 2023-06-10
> > > **Response to review questions 3/n**
> > >
> > > ## Requested changes
> > > > I would definitely like the clarity of the paper to be improved, but even if that were the case, I am actually not sure what it would take to secure my recommendation of acceptance to TMLR. Reading the evaluation criteria, I noticed that there is no place for survey papers (even very clear ones): TMLR seems to want each paper to contain some form of new knowledge. I think the criterion that could best apply to this paper is: accounts of applications of existing techniques that shed light on the strengths and weaknesses of the methods; However, as I mentioned above I don't think that this paper does this. The authors themselves sought to "guide researchers on their research problems involving bi-level optimization", but I don't think this paper really does so, and I don't see an easy fix for this without significantly changing the focus of the paper. So my overall requested change would be "do something which adds new knowledge to the field or explain what new knowledge your paper does contribute which I missed during my review". Some ideas for this are:compare/contrast various approximate bilevel optimization methods in terms of speed, accuracy, and memory discuss pros/cons of implicit vs explicit bilevel optimization methods discuss for which problems it is most suitable to formalize them as bilevel optimization methods. For example, hyperparmeter optimization could be formalized in other ways besides bilevel optimization. It might not be the best tool for every problem.
> > >
> > > Thank you for your valuable and insightful feedback. We acknowledge your concerns regarding the originality and practical applicability of our paper.
> > >
> > > In response, we have made significant additions to our manuscript to present new perspectives, provide comprehensive comparisons, and discuss potential applications of bilevel optimization methods. Specifically, we have included Section 4.5 which offers an in-depth comparative analysis of various approximate bilevel optimization methods. This section assesses these methods based on speed, accuracy, and memory usage and discusses the pros and cons of both implicit and explicit bilevel optimization methods. This part aims to provide guidance to researchers in choosing the appropriate method for their specific problems.
> > >
> > > Further, we have elaborated on the types of problems that are most suited for formulation as bilevel optimization methods. We addressed this in the opening of Section 3.1, highlighting that such problems require the existence of a main optimization component and a differentiable relationship between inner and outer variables: "A particular single-task problem can be deemed suitable for bi-level optimization if it meets two criteria.  Firstly, it has a main optimization problem guiding the optimization of the outer variable. Secondly, a constraint exists between the inner and outer variables such that a differentiable relationship between these variables can be established. To elaborate, our first step is to identify the inner variable, denoted as $\boldsymbol{\theta}$, and the outer variable, $\boldsymbol{\phi}$. Next, we identify the main optimization component which optimizes the hyperparameters, which acts as the outer level problem.  Finally, the inner level problem is framed by recognizing the constraint between these two variables, which further enables us to establish a differentiable relationship between them."

---

> > > > ### Author Response · Authors · 2023-06-10
> > > > **Response to review questions 4/n**
> > > >
> > > > We also expanded on our discussion of DARTS, emphasizing the crucial role of transforming discrete variables into continuous ones, especially for discrete hyperparameter optimization: "It's notable that the conversion of discrete variables to continuous ones holds significant importance in gradient-based bi-level optimization as it offers an efficient method to adjust discrete parameters. This technique finds its application in earlier label learning methodologies, as demonstrated in the works by \citep{algan2021meta, wu2021learning}. In these cases, discrete one-hot labels are morphed into soft labels for optimization. However, in such instances, the conversion to continuous forms isn't as crucial as in our present context, owing to the existence of alternate methods like instance reweighting for managing label noise."
> > > >
> > > > Finally, in the context of high-dimensional regularization parameter optimization, we underscored the advantages of the bilevel optimization framework over traditional methods like random search and Bayesian Optimization: "When dealing with high dimensionalities, traditional methods such as random search and Bayesian Optimization can prove inadequate. In contrast, the bi-level optimization framework offers an efficient approach to directly update high-dimensional hyperparameters, such as regularization, as demonstrated by \citet{rendle2012learning}, \citet{chen2019lambdaopt}, and \citet{lorraine2020optimizing}. "
> > > >
> > > > In conclusion, our revisions strive to present fresh insights, underscore commonalities, and offer practical guidance for future research. We believe these enhancements will address your concerns and contribute new knowledge to the field. We look forward to further feedback.
> > > >
> > > > ## To Conclude
> > > > We hope that our response sufficiently addresses your concerns. We are deeply grateful for your insightful review and the valuable feedback you provided. Your careful examination of our paper has been instrumental to our revision process. We eagerly anticipate further discussions during the rebuttal phase.
> > > >
> > > >     [1] Luca Franceschi, Paolo Frasconi, Saverio Salzo, Riccardo Grazzi, and Massimiliano Pontil. Bilevel programming for hyperparameter optimization and meta-learning. In International Conference on Machine Learning, 2018.

---

> > > > > ### Author Response · Authors · 2023-06-17
> > > > > **Looking forward to your feedback**
> > > > >
> > > > > Dear Reviewer,
> > > > >
> > > > > I hope this message finds you well. I am reaching out in regard to our recent correspondence about our manuscript. To ensure that your feedback has been fully addressed, I would like to summarize the changes and clarifications that we have implemented in our revised version.
> > > > >
> > > > > **Clarity Improvements**
> > > > >
> > > > > 1. For Equations 1-2, we recognized the validity of your observation and removed the specific labels "training" and "validation" from the equations to avoid unnecessary confusion.
> > > > >
> > > > > 2. To clarify the term "meta knowledge", we emphasized its synonymity with "meta-learning" in the context of our paper. This term is now more clearly explained in Section 2.
> > > > >
> > > > > 3. Considering your feedback on the presentation of Von Stackelberg games, we revised our manuscript to limit mentions of this concept prior to its full explanation in Section 3.1.2.
> > > > >
> > > > > 4. In response to your comment about the brief descriptions of the surveyed methods, we expanded upon these methods and clarified how each one is formulated within the framework of bilevel optimization.
> > > > >
> > > > > **Enhanced Insights and New Knowledge**
> > > > >
> > > > > 1. To tackle your concern about the lack of unifying insights, we made substantial revisions to Section 3, with a more structured presentation, enhanced readability, and inclusion of our unique insights to highlight the commonalities across different works.
> > > > >
> > > > > 2. In light of your feedback on the key challenges in bilevel optimization, we added Section 4.5 to discuss the trade-off between memory usage, computation time, and accuracy.
> > > > >
> > > > > 3. We introduced substantial new content to meet the standard of TMLR for new knowledge. This includes an in-depth comparative analysis of various approximate bilevel optimization methods (Section 4.5), an expanded discussion on the types of problems suitable for formulation as bilevel optimization methods, and an in-depth discussion of DARTS and its crucial role in transforming discrete variables into continuous ones.
> > > > >
> > > > > We are grateful for your thorough review and constructive comments, which have greatly assisted us in enhancing the quality and clarity of our work. We look forward to hearing from you regarding these revisions.
> > > > >
> > > > > Best Regards,
> > > > >
> > > > > Paper 1065 Authors.

---

> > > > > > ### Comment · Reviewer_UciL · 2023-06-21
> > > > > > **Appreciate revisions but still find the paper unclear and not very insightful**
> > > > > >
> > > > > > Dear authors,
> > > > > >
> > > > > > Thank you very much for the revisions. I have re-read the paper and thought about the changes you have made. I think that the paper is improved, but unfortunately I am still not enthusiastic about the paper. My thoughts are below.
> > > > > >
> > > > > > - *meta knowledge*: I appreciate that you have clarified your intended usage of this term, but I still think it is non-standard and confusing. "meta-knowledge" is knowledge about knowledge, just like "meta learning" is learning to learn or "meta-theory" is theory about theory.
> > > > > > - I still think the taxonomy in section 3 is awkward and not particularly insightful. I think the definition of "explicit hypergradient" is not made clear until section 4 when the chain rule is applied. Classifying the methods based on "same vs different formula" and "explicit / implicit hypergradient" still seemed odd to me. I did not see why this was a useful perspective to understand different methods. Same with "single-task" vs "multi-task": although bilevel optimization is used in these different settings I don't think they are fundamentally very different: in each case the training datasets are just sampled from different distributions, but the loss formulas can be very similar.
> > > > > > - Even accepting this taxonomy, some of the methods seemed incorrectly classified. For example, methods a-c in Table 2 don't actually seem to have the same formula for the inner and outer objective. Is this a mistake or am I missing something?
> > > > > > - While I appreciated the analysis in section 4.5, I think the focus on scaling missed out on the fact that many of these methods are *approximate*, so there is also an accuracy trade-off which is harder to characterize.
> > > > > > - I did not see how the content in section 5 relates to gradient-based bilevel optimization: although this could be used for all of these problems, it is not clear that it is a particularly promising approach. If the authors see a promising research agenda here I think it would need to be explained in more detail
> > > > > > - The revised paper is very long: almost 20 pages. I think it would be better if the paper were shorter. I think the key aims of the paper could be achieved in under 12 pages. Perhaps extra content could be moved to the appendix?
> > > > > > - I think the guidance for formulating problems as bi-level optimization objectives on page 3 is not very comprehensive or helpful. It essentially says "if your problem has two equations which are coupled where one is constrained on the other then it can be written as a bilevel optimization problem". I think that is close to a tautology.

---

> > > > > > > ### Author Response · Authors · 2023-06-23
> > > > > > > **Response to review questions 1/n**
> > > > > > >
> > > > > > > > Thank you very much for the revisions. I have re-read the paper and thought about the changes you have made. I think that the paper is improved, but unfortunately I am still not enthusiastic about the paper. My thoughts are below.
> > > > > > >
> > > > > > > Thank you once again for dedicating your time to review our paper and offering your thoughtful comments. We understand that despite our efforts in revising the manuscript, you still have reservations about its content. Below, we aim to address your ongoing concerns.
> > > > > > >
> > > > > > > > meta knowledge: I appreciate that you have clarified your intended usage of this term, but I still think it is non-standard and confusing. "meta-knowledge" is knowledge about knowledge, just like "meta learning" is learning to learn or "meta-theory" is theory about theory.
> > > > > > >
> > > > > > > Thank you for your observation regarding our usage of the term "meta-knowledge." Our intention behind using this term was to adhere to its usage in the context of meta-learning, where it is used to describe knowledge that is transferrable across multiple tasks. This usage of "meta-knowledge" is derived directly from the meta-learning survey paper by [2], wherein "meta-knowledge" signifies insights acquired across various tasks rather than knowledge confined to a singular task.
> > > > > > >
> > > > > > >
> > > > > > > > I still think the taxonomy in section 3 is awkward and not particularly insightful. I think the definition of "explicit hypergradient" is not made clear until section 4 when the chain rule is applied. Classifying the methods based on "same vs different formula" and "explicit / implicit hypergradient" still seemed odd to me. I did not see why this was a useful perspective to understand different methods. Same with "single-task" vs "multi-task": although bilevel optimization is used in these different settings I don't think they are fundamentally very different:
> > > > > > >
> > > > > > > We appreciate your continued engagement with our paper and the observations you have shared.
> > > > > > >
> > > > > > > Regarding the "explicit hypergradient," we agree that introducing it earlier in the paper might help with comprehensibility. As you suggested, we have now moved the explanation of this concept, including the application of the chain rule, to Section 2.
> > > > > > >
> > > > > > > As for the taxonomy, we first divide the discussion based on the distinction between hyperparameter optimization (Section 3.1) and meta-knowledge extraction (Section 3.2). This distinction corresponds to single-task and multi-task formulations respectively. This organization scheme draws from the taxonomy proposed in paper [1]. Our goal here is to provide a clear understanding of the landscape to researchers who may be well-versed in one area (hyperparameter optimization or meta-learning), but not as familiar with the other. By framing both these topics within the context of bi-level optimization, we illustrate their interconnectedness and reveal the potential for their unification.
> > > > > > >
> > > > > > > In addition to this overarching distinction, within the context of single-task formulations, we further categorize based on two criteria:
> > > > > > >
> > > > > > > 1) Whether the inner and outer level loss functions share the same mathematical formula, and
> > > > > > > 2) Whether the hypergradient is only derived from the inner level connection.
> > > > > > >
> > > > > > > These criteria are selected with the intention to guide researchers on how they could formulate their problems as bi-level optimization challenges. Certain scenarios, such as a shared mathematical formula with an explicit hypergradient or differing mathematical formulas without an explicit hypergradient, might not be immediately intuitive to readers. By providing a structured guide, we aim to help researchers to frame their unique bi-level optimization problems effectively. We have provided additional explanations for these classification criteria at the beginning of Section 3.1 for better clarity as below.
> > > > > > >
> > > > > > > "(1) A notable situation is when the constraint and the main optimization problem use the same mathematical formula, yet they approach the problem from two different viewpoints. For instance, they might utilize the same mathematical formula, but one focuses on validation loss while the other focuses on training loss. However, we also consider scenarios where the main optimization problem and the constraint use entirely different mathematical formulas, implying that they optimize entirely different problems. This comprises the first criterion for our evaluation. (2) In some cases, the main optimization problem might not directly contain the outer variable. In such cases, the connection built at the inner level is utilized. This situation poses a challenge in formulating the outer level task, thereby leading us to introduce a second criterion. This second criterion classifies works based on whether the calculation of the hypergradient relies exclusively on the established inner level connection"
> > > > > > >
> > > > > > > We understand your concerns and will make it a point to clarify the benefits of this classification scheme within the paper.

---

> > > > > > > > ### Author Response · Authors · 2023-06-23
> > > > > > > > **Response to review questions 2/n**
> > > > > > > >
> > > > > > > > > in each case the training datasets are just sampled from different distributions, but the loss formulas can be very similar.
> > > > > > > >
> > > > > > > >
> > > > > > > > We agree with your observation that for many instances in bi-level optimization, the inner and outer level loss functions can be similar, especially when the training datasets are samples from different distributions. However, this is not always the case. The beauty of bi-level optimization lies in its adaptability to a variety of scenarios, including ones where the inner and outer level losses are entirely distinct.
> > > > > > > >
> > > > > > > > To illustrate, consider the example of a topology design problem. Once a topology (represented by $\phi$) is defined, the system converges to an equilibrium state, $\theta({\phi})$. This state is achieved by minimizing an energy function, which serves as our inner level loss function. Once equilibrium is achieved, the system cost, represented by ${L}^{out}({\theta}({\phi}))$, can be calculated. This system cost serves as our outer level loss function. Hence, we formulate this as a bi-level optimization problem where the inner level loss (the energy function) and the outer level loss (the system cost) are fundamentally different from each other.
> > > > > > > >
> > > > > > > > This diversity in the nature of the loss functions underlines the importance and relevance of our proposed taxonomy, as it caters to scenarios where the loss formulas could be either similar or starkly different.
> > > > > > > >
> > > > > > > >
> > > > > > > >
> > > > > > > > > Even accepting this taxonomy, some of the methods seemed incorrectly classified. For example, methods a-c in Table 2 don't actually seem to have the same formula for the inner and outer objective. Is this a mistake or am I missing something?
> > > > > > > >
> > > > > > > > We apologize for the confusion and appreciate your valuable feedback. We classify the methods in Table 2 as having the same formula for the inner and outer objective because the primary objective being optimized remains consistent across both levels. There may be minor variations in the representation of this objective, but these do not fundamentally alter the mathematical form of the problem.
> > > > > > > >
> > > > > > > > For instance:(a) For the formulation illustrated in Table 2 (a), in the revised version, we have clarified by adding the sentence, "In the outer level loss, the regularization term is considered as zero, making it the same as the inner level loss in mathematical form.";(b) Regarding the formulation in Table 2 (b), we've added the clarification, "Though the inner level problem is represented through the perspective of an optimizer, fundamentally it's still the same math objective being optimized as in the outer level loss."; (c) For the formulation in Table 2 (c), we've specified in the revised version by stating, "The outer level loss, with zero perturbation, can be viewed as being the same math form as the inner level loss."
> > > > > > > >
> > > > > > > > An example where inner and outer level losses are distinct can be seen in the context of the topology design problem. When a topology, $\phi$, is provided, the system reaches an equilibrium state, $\theta({\phi})$, by minimizing the energy function (inner level loss). Following this, the system cost, ${L}^{out}({\theta}({\phi}))$, is computed which serves as the outer level loss. In this instance, the inner level energy function and the outer level system cost symbolize entirely different objectives, making the losses fundamentally different.
> > > > > > > >
> > > > > > > >
> > > > > > > > > While I appreciated the analysis in section 4.5, I think the focus on scaling missed out on the fact that many of these methods are approximate, so there is also an accuracy trade-off which is harder to characterize.
> > > > > > > >
> > > > > > > > We agree with your point. The accuracy of solutions obtained from these methods indeed matters, alongside the computational efficiency. In the context of bi-level optimization, accuracy pertains to how well the method solves the inner-level problem, which directly influences the quality of the outer-level solution.
> > > > > > > >
> > > > > > > > In Section 4.5, we mainly focus on the computational cost (scaling) of different methods due to its clear and measurable characteristics. The trade-off between computational efficiency and solution accuracy is complex and case-dependent, which makes it a challenging aspect to fully analyze and generalize.
> > > > > > > >
> > > > > > > > For instance, while the explicit proxy update method usually provides less accurate solutions compared to the explicit gradient update and the implicit gradient update, it may still be preferred in some cases due to its lower computational cost. Conversely, when accuracy is prioritized, one might choose the explicit gradient update or the implicit gradient update despite their higher computational costs.
> > > > > > > >
> > > > > > > > Your comment reminds us of the importance of addressing this trade-off, and we will make an effort to emphasize this point more prominently in our future work. We thank you for this insightful feedback.

---

> > > > > > > > > ### Author Response · Authors · 2023-06-23
> > > > > > > > > **Response to review questions 3/n**
> > > > > > > > >
> > > > > > > > > > I did not see how the content in section 5 relates to gradient-based bilevel optimization: although this could be used for all of these problems, it is not clear that it is a particularly promising approach. If the authors see a promising research agenda here I think it would need to be explained in more detail
> > > > > > > > >
> > > > > > > > > We appreciate your feedback on Section 5 of our paper. The primary purpose of this section is to envision potential future research avenues for gradient-based bi-level optimization.
> > > > > > > > >
> > > > > > > > > The first subsection, "Effective Data Optimization for Science", explores how gradient-based bi-level optimization can be applied to optimize tuning hyperparameters, design parameters, and feature selection in scientific domains. Each of these data types serves as the outer variable in the bi-level optimization problem and can be updated effectively via gradient optimization to help solve scientific challenges.
> > > > > > > > >
> > > > > > > > > In the scientific domain, there is a vast array of data types and problems that could potentially benefit from bi-level optimization. Examples mentioned in our work, such as tuning hyperparameters, design parameters, or learned features, all correspond to at least one paper discussed in the task formulation section. Given the complexity and volume of data in scientific research, we believe gradient-based bi-level optimization has the potential to significantly contribute to data optimization efforts in this area.
> > > > > > > > >
> > > > > > > > > The second subsection, "Accurate Explicit Proxy Update", proposes a future research agenda aimed at constructing a more accurate proxy in explicit proxy update. This research direction could further improve the effectiveness and accuracy of bi-level optimization algorithms, thereby broadening their applicability and performance in various tasks.
> > > > > > > > >
> > > > > > > > >
> > > > > > > > >
> > > > > > > > > > The revised paper is very long: almost 20 pages. I think it would be better if the paper were shorter. I think the key aims of the paper could be achieved in under 12 pages. Perhaps extra content could be moved to the appendix?
> > > > > > > > >
> > > > > > > > > We understand your concern about the length of the paper, and we appreciate your suggestion. In response, we will move some of the more detailed formulation content to the appendix in order to streamline the main body of the paper.
> > > > > > > > >
> > > > > > > > > However, it's worth noting that for a comprehensive survey paper such as this one, a length of around 20 pages is quite common. This allows us to cover the breadth and depth of the field, providing a holistic view to the readers.
> > > > > > > > >
> > > > > > > > > > I think the guidance for formulating problems as bi-level optimization objectives on page 3 is not very comprehensive or helpful. It essentially says "if your problem has two equations which are coupled where one is constrained on the other then it can be written as a bilevel optimization problem". I think that is close to a tautology.
> > > > > > > > >
> > > > > > > > > Our apologies for any misunderstanding. We want to emphasize that the process of formulating problems as bi-level optimization objectives is more nuanced than simply recognizing a problem consists of two coupled equations.
> > > > > > > > >
> > > > > > > > > The essence of this guidance is to direct researchers to first consider the practical aspects of the problem at hand: identify what is the outer variable that you are trying to optimize (the outer problem) and what constraints are affecting this objective (the inner problem).
> > > > > > > > >
> > > > > > > > > Two coupled equations don't inherently constitute a bi-level optimization problem unless one equation can be viewed as the constraint of the other with a real-world meaning and the constraint equation can be differentiated to establish a clear relationship between the outer variable (what we aim to optimize) and the inner variable. We aim to illuminate that identifying this differentiable relationship between the inner and outer variables is essential to compute the hypergradient, a key aspect of gradient-based bi-level optimization. Not every coupled equation can provide this differentiable relationship, thus it's crucial to consider these aspects while formulating gradient-based bi-level optimization problems.
> > > > > > > > >
> > > > > > > > >
> > > > > > > > >     [1] Luca Franceschi, Paolo Frasconi, Saverio Salzo, Riccardo Grazzi, and Massimiliano Pontil. Bilevel programming for hyperparameter optimization and meta-learning. In International Conference on Machine Learning, 2018.
> > > > > > > > >
> > > > > > > > >     [2] Hospedales T, Antoniou A, Micaelli P, et al. Meta-learning in neural networks: A survey[J]. IEEE transactions on pattern analysis and machine intelligence, 2021, 44(9): 5149-5169.

---

### Review · Reviewer_LFqX · 2023-05-29

**Summary Of Contributions:**

This paper provides a survey of gradient-based bi-level optimization for deep learning.  It defines gradient-based bi-level optimization and then covers four paradigms for a setting of a single task and two paradigms for a setting of multi tasks.  After that, the authors introduce some optimization strategies for the topics of this work.  Eventually, future direction and conclusion are described.

**Audience:**

Yes

**Broader Impact Concerns:**

I do not have any broader impact concerns.

**Claims And Evidence:**

No

**Requested Changes:**

To better this paper, the following points should be updated:

* The taxonomy of gradient-based bi-level optimization should be improved more clearly.

* Description of each category should be covered in a consolidated manner.

* Discussion on challenging issues and future directions should be enhanced.

* Writing and presentation should be improved.

### Minor Issues

* Every equation is a sentence.  A comma or period should be used appropriately.

* Mathematical and regular fonts should be used appropriately.

* I am not sure this sentence "In this paper, the hyperparameters are not limited to the regularization and the learning rate but refer to any knowledge in a single task formulation including network architecture, and distilled data samples, as we will illustrate more detailedly in Section 3." is required.  Network architecture and distilled data samples are obviously hyperparameters.

* I am not sure that this sentence "The single-task formulation applies bi-level optimization on a single task and aims to learn hyperparameters for the task." is correct.  It is only related to hyperparameter optimization?

* Figure 1 is not about the summary of gradient-based bi-level optimization; it is now about the contents of this paper.  It should be fixed by covering the taxonomy of gradient-based bi-level optimization appropriately.

* \citep and \citet should be used appropriately.

**Strengths And Weaknesses:**

### Strengths

* Gradient-based bi-level optimization is an important topic in a machine learning field.

### Weaknesses

* I think that the structure of this survey paper is not well-organized.  For example, although the number of tasks is one of criteria to classify strategies for the gradient-based bi-level optimization, there might be similar strategies in both single-task and multi-task settings.  It should be covered in an integrated way.

* Following the above concern, the taxonomy of gradient-based bi-level optimization should be revised.  I think that the current taxonomy is too simple and not clear.

* Writing and presentation can be improved more.

* Discussion on future directions is weak.  Since this paper is a survey paper, it should be carefully discussed.

---

> ### Author Response · Authors · 2023-06-10
> **Response to review questions 1/n**
>
> ## General Reply
>
> Thank you for taking the time to provide us with your valuable and constructive comments. We appreciate your feedback and insights and we have carefully revised the paper according to your comments, as outlined below.
>
> ## Weakness
>
>
> > I think that the structure of this survey paper is not well-organized. For example, although the number of tasks is one of criteria to classify strategies for the gradient-based bi-level optimization, there might be similar strategies in both single-task and multi-task settings. It should be covered in an integrated way. Following the above concern, the taxonomy of gradient-based bi-level optimization should be revised. I think that the current taxonomy is too simple and not clear.
>
> We thank you for your feedback concerning the organization of our survey paper, particularly the taxonomy of gradient-based bi-level optimization strategies.
>
> In our classification of bi-level optimization tasks into single-task and multi-task categories, we extend beyond merely the number of tasks. This division is more fundamentally rooted in the different objectives and types of knowledge targeted by each setting. The single-task formulation primarily focuses on obtaining task-specific knowledge through hyperparameter tuning, while the multi-task formulation seeks to derive meta knowledge applicable across multiple tasks. This classification criterion, adopted from paper [1], is intended to give a clear overview to researchers in both the hyperparameter optimization and meta-learning fields, also demonstrating their unification potential through bi-level optimization. This allows researchers unfamiliar with one area to get a clear overview of both, highlighting their interconnection.
>
> Indeed, similar strategies can be found across both settings, but they hold different meanings in different contexts. For example, a regularization parameter in a single-task setting serves to control overfitting and is task-specific. Conversely, in a multi-task setting, a learned optimizer aims to optimize task loss, representing meta knowledge acquired from different tasks.
>
> In response to your feedback, we've elaborated on task formulation in the beginning of Section 3, Section 3.1 and Section 3.2 of the revised paper to further clarify our classification rationale with blue text. This clarification should make it more evident that, while some methods may appear similar between single-task and multi-task settings, their applications and the knowledge they encapsulate differ significantly. We believe this refines the structure of our survey and offers a more nuanced understanding of gradient-based bi-level optimization strategies.
>
> > Writing and presentation can be improved more.
>
> We appreciate your feedback and have worked diligently to enhance both the writing and presentation of our paper in this revised version with the blue font. This includes addressing your suggestions regarding minor issues.

---

> > ### Author Response · Authors · 2023-06-10
> > **Response to review questions 2/n**
> >
> > > Discussion on future directions is weak. Since this paper is a survey paper, it should be carefully discussed.
> >
> > We have incorporated Section 5.2 to provide an in-depth discourse on potential future work. The additional text can be found below.
> >
> > "As we conclude our survey, we spotlight an emerging domain in gradient-based bi-level optimization as a future direction: **Effective Data Optimization for Science**. Depending on the nature of the data, this domain can be primarily divided into three areas: tuning parameter optimization, design optimization, and feature optimization. Here, we elaborate on how bi-level optimization acts as a potent tool to optimize these categories of data, aiding in solving scientific problems.
> >
> > **Tuning parameter optimization.** In the scientific domain, a common problem is to find a design - be it a protein, a material, a robot, or a DNA sequence - that optimizes a specific black-box objective function such as a protein property score[^trabucco2022design]. This process often trains a proxy model using collected pairs of designs and scores along with certain tuning parameters, and the trained proxy guides the design search in evolutionary algorithms[^hansen2006cma] or reinforcement learning[^angermueller2019model]. Parameter tuning via bi-level optimization has emerged as a promising approach to achieve data-specific optimization and can enhance the local accuracy of the proxy model around the current point in the search space. This could potentially revolutionize the design search process, making it more effective and targeted. Such optimization can be achieved by locally sampling data points from the collected design-score pairs or by using pseudo-labelers to label neighboring samples. As mentioned earlier, we envision utilizing a clean validation set to fine-tune these data-specific parameters via bi-level optimization. This approach offers promising prospects in identifying reliable portions of locally sampled data, thereby improving the local accuracy of the proxy model, and providing a more effective direction to the search process. As we move forward, the nuances of these techniques will be refined and expanded, offering greater possibilities in design optimization tasks.
> >
> > **Design optimization.** A paradigm shift in approaching scientific problems could be to focus on directly optimizing the design, instead of tuning parameters to enhance proxy model performance. Data distillation through bi-level optimization could serve as a pivotal strategy in this context. This technique extracts extensive knowledge from large datasets into distilled samples, a process which has been found to retain features corresponding to the labels associated with the distilled samples[^wang2018dataset][^lei2023comprehensive]. Taking inspiration from these findings, certain studies assign a predefined high score to an initial design and distill knowledge from the training set into the design[^chen2022bidirectional][^chen2023bidirectional]. This procedure results in a design with high-scoring features, essentially creating a desirable final candidate. As we look towards the future of gradient-based bi-level optimization, two intriguing directions arise from this research line. Firstly, employing foundational models specific to fields such as materials science, alongside advanced data distillation techniques[^lei2023comprehensive], could substantially improve distillation performance. This would enable more accurate high-scoring features to be incorporated into the design. Secondly, analyzing the distilled high-scoring sample to extract patterns and potential rules presents another exciting prospect. Armed with this derived knowledge, it may be possible to create high-scoring designs or to integrate this understanding into our modeling process, paving the way for more insightful and accurate predictions. (to be continued)"

---

> > > ### Author Response · Authors · 2023-06-10
> > > **Response to review questions 3/n**
> > >
> > > "**Feature optimization.** Another pivotal area in science is feature optimization, which aims to improve the learning of data features. This optimization process is intertwined with scientific constraints, which can be effectively incorporated as the inner-level task in bi-level optimization. These constraints generally include geometric and biochemical constraints. Geometric constraints stem from the physical or spatial characteristics of a system. An example of leveraging these constraints can be seen in the work of[^xu2021end]. Here, the process of molecular conformation prediction is decomposed into two levels, with the geometric constraint between molecular conformation and predicted atom distances being the inner level task. By utilizing this constraint, the trained neural network can produce atom distance features that are more compatible with the actual molecular structure. Biochemical constraints are rooted in the chemical and biological principles that govern a system. These constraints have been utilized by[^chen2022structure], where the inherent correspondence between the sequential and structural representations of a protein is used as the biochemical constraint. The task of maximizing mutual information is formulated as the inner-level task, which allows the Graph Neural Network (GNN) to output protein structural features with higher accuracy. Moving forward, feature optimization is set to play an increasingly important role in enhancing the learning of data features by leveraging inherent scientific constraints within a bi-level optimization framework.
> > >
> > > Beyond these opportunities, another important future direction lies in devising proficient acceleration strategies. This direction is aimed at enhancing the performance of hypergradient computation. A promising intersection lies between explicit gradient update and explicit proxy update, which could borrow accurate computation from the former and efficiency from the latter. This approach presents the possibility of more efficient hypergradient computation in gradient-based bi-level optimization."
> > >
> > > ## Minor Issues
> > >
> > > > Every equation is a sentence. A comma or period should be used appropriately. Mathematical and regular fonts should be used appropriately. \citep and \citet should be used appropriately.
> > >
> > > We concur with your suggestion and have thoroughly revised the manuscript accordingly. We have now ensured that each equation is treated as a sentence with appropriate punctuation. Moreover, we have taken special care to distinguish between the mathematical and regular fonts in our text. The usage of \citet and \citep has also been reviewed and adjusted to fit the context appropriately. We appreciate your attention to detail and we believe these modifications have improved the clarity and readability of our manuscript.
> > >
> > >
> > > > I am not sure this sentence "In this paper, the hyperparameters are not limited to the regularization and the learning rate but refer to any knowledge in a single task formulation including network architecture, and distilled data samples, as we will illustrate more detailedly in Section 3." is required. Network architecture and distilled data samples are obviously hyperparameters.
> > >
> > > While traditional perspectives often limit the scope of hyperparameters to factors such as regularization and learning rate, we acknowledge that readers from diverse backgrounds may interpret this differently and possibly restrict their understanding to these traditional parameters, which can potentially lead to misconceptions. To improve clarity and accuracy, we have revised our statement: "In this paper, the hyperparameters are not limited to the regularization and the learning rate but refer to any knowledge in a single task formulation, as we will illustrate more detailedly in Section 3."
> > >
> > > Moreover, in certain single-task formulations, the outer variable doesn't necessarily represent "hyperparameters". For instance, in the Stackelberg game, the outer variable could denote the number of products $\phi$ of the leading company, and in molecular conformation prediction [2], it corresponds to the atom distance prediction network $\phi$.

---

> > > > ### Author Response · Authors · 2023-06-10
> > > > **Response to review questions 4/n**
> > > >
> > > > > I am not sure that this sentence "The single-task formulation applies bi-level optimization on a single task and aims to learn hyperparameters for the task." is correct. It is only related to hyperparameter optimization?
> > > >
> > > > We acknowledge the feedback on the interpretation of the statement "The single-task formulation applies bi-level optimization on a single task and aims to learn hyperparameters for the task."  To clarify, our usage of the term "hyperparameters" extends beyond the conventional understanding, encapsulating any form of knowledge in a single task formulation. This broader definition is not confined to traditional hyperparameters. As previously discussed, in this context, hyperparameters might include, but are not limited to, elements like network architecture, distilled data samples, or even the number of products in a business model. We hope this provides a clearer picture of our perspective.
> > > >
> > > > > Figure 1 is not about the summary of gradient-based bi-level optimization; it is now about the contents of this paper. It should be fixed by covering the taxonomy of gradient-based bi-level optimization appropriately.
> > > >
> > > > We appreciate your feedback on the structure of our survey paper and your observation about Figure 1.  We would like to clarify that the taxonomy adopted in our paper was intentionally designed to cater to researchers from both hyperparameter optimization and meta-learning communities, providing them with a clear review of their respective fields and illustrating their potential for unification through bi-level optimization.
> > > >
> > > > In response to your earlier feedback, we expanded on the classification rationale in the beginning of Section 3, Section 3.1, and Section 3.2 of the revised paper. We explained that while some methods may appear similar between single-task and multi-task settings, their applications and the knowledge they encapsulate differ significantly due to their distinctive objectives.
> > > >
> > > > To further elaborate, we have added text at the beginning of Section 3.1 to provide a more in-depth explanation of our single-task taxonomy:
> > > >
> > > > "(1). A notable situation is when the constraint and the primary optimization problem use the same mathematical formula, yet they approach the problem from two different viewpoints. For instance, they might utilize the same mathematical formula, but one focuses on validation loss while the other focuses on training loss. However, we also consider scenarios where the primary optimization problem and the constraint use entirely different mathematical formulas, implying that they optimize entirely different problems. This comprises the first criterion for our evaluation. (2). In some cases, the main optimization problem might not directly contain the outer variable. In such cases, the connection built at the inner level is utilized. This situation poses a challenge in formulating the outer level task, thereby leading us to introduce a second criterion. This second criterion classifies works based on whether the calculation of the hypergradient relies exclusively on the established inner level connection."
> > > >
> > > > We hope these changes address your concerns and make the structure of the paper, as well as the reasoning behind our taxonomy, more clear.
> > > >
> > > > ## Overall
> > > > Does the above reply address your concerns? Thank you again for your instructive review and feedback. We very much appreciate your careful review of the paper and look forward to further exchange with you during the rebuttal phase.
> > > >
> > > >     [1] Luca Franceschi, Paolo Frasconi, Saverio Salzo, Riccardo Grazzi, and Massimiliano Pontil. Bilevel programming for hyperparameter optimization and meta-learning. In International Conference on Machine Learning, 2018.
> > > >
> > > >     [2] Minkai Xu, Wujie Wang, Shitong Luo, Chence Shi,Yoshua Bengio, Rafael Gomez-Bombarelli, and Jian Tang. An end-to-end framework for molecular conformation generation via bilevel programming. In International Conference on Machine Learning, 2021

---

> > > > > ### Author Response · Authors · 2023-06-17
> > > > > **Looking forward to your feedback**
> > > > >
> > > > > Dear Reviewer,
> > > > >
> > > > > I hope this message finds you well. I wanted to follow up on the revisions we've made to our paper in response to your insightful comments and recommendations. We greatly appreciate the time and effort you invested in reviewing our work. As such, we have made several adjustments to address each of your concerns. Below, I have listed the main revisions we've made based on your feedback:
> > > > >
> > > > > 1. Regarding the organization of the survey paper: We have clarified the classification of bi-level optimization tasks into single-task and multi-task categories. We've expanded our discussion on this in the beginning of Section 3, Section 3.1 and Section 3.2, to elaborate our rationale. We hope this will provide a more nuanced understanding of gradient-based bi-level optimization strategies.
> > > > >
> > > > > 2. Writing and presentation: We have enhanced both the writing and presentation of our paper. We also paid careful attention to the minor issues you pointed out, such as punctuation after equations, appropriate use of mathematical and regular fonts, and the correct usage of citation commands.
> > > > >
> > > > > 3. Future directions: We have expanded our discussion on potential future work in Section 5.2. We've highlighted several emerging areas in gradient-based bi-level optimization that hold promise for future exploration, including tuning parameter optimization, design optimization, and feature optimization.
> > > > >
> > > > > 4. Use of the term 'hyperparameters': We have clarified that our use of the term "hyperparameters" extends beyond the conventional understanding to include any knowledge in a single task formulation.
> > > > >
> > > > > 5. Figure 1: In response to your feedback, we have further expanded the discussion on our single-task taxonomy at the beginning of  Section 3.1, providing a more in-depth explanation of our reasoning behind the taxonomy.
> > > > >
> > > > > Once again, we appreciate your valuable feedback. If there are any additional questions or comments, we look forward to addressing them in the rebuttal phase. We believe these revisions have addressed the concerns you raised, and we hope they make our work clearer and more robust.
> > > > >
> > > > > Thank you for your time and consideration.
> > > > >
> > > > > Best regards,
> > > > >
> > > > > Paper 1065 Authors.

---

> > > > > > ### Comment · Reviewer_LFqX · 2023-06-18
> > > > > > **Reply to the authors' comments and revision**
> > > > > >
> > > > > > Thank you for your comments!
> > > > > >
> > > > > > I have read a new revision as well as your comments.
> > > > > >
> > > > > > Some changes make a manuscript much clearer.
> > > > > >
> > > > > > However, I still have concerns on the structure of this paper and future directions.
> > > > > >
> > > > > > Although the submission is revised, the taxonomy of gradient-based bi-level optimization is still unclear.
> > > > > >
> > > > > > In addition, I think that the future directions are not constructive.  I agree that AI for science is an important direction of gradient-based bi-level optimization and also machine learning.  However, it is not all we anticipate now.
> > > > > >
> > > > > > I am happy to hear any thoughts on my concerns and I am going to discuss these with other reviewers and an action editor.

---

> > > > > > > ### Author Response · Authors · 2023-06-18
> > > > > > > **Rebuttal to Review Feedback: Clarification on Taxonomy**
> > > > > > >
> > > > > > > Thank you for your continued engagement and feedback!
> > > > > > >
> > > > > > > Regarding your concerns on the paper's structure and the taxonomy of gradient-based bi-level optimization, we have striven to provide clarity and structure in the following manner:
> > > > > > >
> > > > > > > Initially, we have divided our subject matter into two main categories: hyperparameter optimization, which corresponds to the single-task formulation discussed in Section 3.1, and meta-knowledge extraction, which aligns with the multi-task formulation covered in Section 3.2.
> > > > > > >
> > > > > > > The rationale behind this classification, as inspired by paper [1], is to provide a comprehensive and clear overview to researchers in the domains of hyperparameter optimization and meta-learning. Our goal is to highlight the potential unification of these areas via bi-level optimization and provide researchers unfamiliar with either area an easy way to grasp their interconnections.
> > > > > > >
> > > > > > > Within the single-task formulation, we've further applied two criteria: (1) whether the inner level loss and the outer level loss share the same mathematical formula; (2) whether the hypergradient only originates from the inner level connection. The intention behind these criteria is to assist researchers in formulating their research problem as a bi-level optimization problem. This is especially useful for non-intuitive cases, such as when the same mathematical formula yields an explicit hypergradient, or when different mathematical formulas do not result in an explicit hypergradient. We trust this classification will be of great use to researchers in formulating their own bi-level optimization problem, and we've provided further details on this as below (also in the beginning of Section 3.1).
> > > > > > >
> > > > > > > > A particular single-task problem can be deemed suitable for bi-level optimization if it meets two criteria. Firstly, it has a main optimization problem guiding the optimization of the outer variable.Secondly, a constraint exists between the inner and outer variables such that a differentiable relationship between these variables can be established.
> > > > > > > To elaborate, our first step is to identify the inner variable, denoted as $\boldsymbol{\theta}$, and the outer variable, $\boldsymbol{\phi}$. Next, we identify the main optimization component which optimizes the hyperparameters, which acts as the outer level problem.  Finally, the inner level problem is framed by recognizing the constraint between these two variables, which further enables us to establish a differentiable relationship between them.
> > > > > > > (1) A notable situation is when the constraint and the main optimization problem use the same mathematical formula, yet they approach the problem from two different viewpoints. For instance, they might utilize the same mathematical formula, but one focuses on validation loss while the other focuses on training loss. However, we also consider scenarios where the main optimization problem and the constraint use entirely different mathematical formulas, implying that they optimize entirely different problems. This comprises the first criterion for our evaluation. (2) In some cases, the main optimization problem might not directly contain the outer variable. In such cases, the connection built at the inner level is utilized. This situation poses a challenge in formulating the outer level task, thereby leading us to introduce a second criterion. This second criterion classifies works based on whether the calculation of the hypergradient relies exclusively on the established inner level connection.
> > > > > > >
> > > > > > > In terms of the multi-task formulation, we have classified the topics based on the two main types of meta-knowledge: model initialization and optimizer.

---

> > > > > > > > ### Author Response · Authors · 2023-06-18
> > > > > > > > **Rebuttal to Review Feedback: Expansion of Future Directions**
> > > > > > > >
> > > > > > > > As for your concerns regarding future directions, the section 'AI for Science' is connected to the task formulation discussed in Section 3. To balance this, we have added another subsection, 'Accurate Explicit Proxy Update,' which corresponds to the optimization techniques discussed in Section 4. This newly added subsection emphasizes the potential of achieving a more efficient and accurate computation of hypergradients by enhancing the explicit proxy update process, as detailed below.
> > > > > > > >
> > > > > > > > ### Future Direction 2: Accurate Explicit Proxy Update
> > > > > > > > Beyond the potential identified in the first future direction, another significant area of exploration centers around constructing a more accurate proxy in explicit proxy update. This promises to yield a more accurate computation of hypergradients efficiently. One viable strategy is to employ a more interpretable construction of the proxy network, $\boldsymbol{\theta}^{*} = P_{\alpha}(\boldsymbol{\phi})$.  For example, \citet{bohdal2021evograd} design a proxy by utilizing a weighted average of the perturbed inner variable, a process that offers interpretability and improved accuracy.
> > > > > > > >
> > > > > > > > Moreover, we underline the connection between model-based optimization~\citep{trabucco2022design} and explicit proxy update, in the context of computing the inner variable from the outer variable using gradient methods.Specifically, the explicit proxy update aims to discover a proxy network that maps the outer variable to the inner variable. This mapping bears a resemblance to the trained model in model-based optimization, which maps the input design (e.g., a robot) to its property (e.g., robot speed). Consequently, recent advancements in model-based optimization can potentially enhance the performance of explicit proxy update. Here, we discuss some directions that are not exhaustive:
> > > > > > > >
> > > > > > > >
> > > > > > > > **Modeling Priors.** In model-based optimization, integrating various modeling priors, such as smoothness~\citep{yu2021roma}, is critical for model performance. Analogously, these priors could be integrated into the training of the proxy network, leading to more precise prediction from the outer to the inner variable.
> > > > > > > >
> > > > > > > >
> > > > > > > >
> > > > > > > > **Importance Sampling.**  Model-based optimization utilizes data gathered during the optimization process, and the model might not be accurate for the neighborhood of the current optimization point. A technique used in model-based optimization~\citep{fannjiang2020autofocused} involves employing importance sampling to retrain the model, based on input distribution. This approach could be adapted to the training of proxy neural network on the already gathered outer-inner variable pairs.
> > > > > > > >
> > > > > > > >
> > > > > > > > **Reverse Mapping.** Some works in model-based optimization ~\citep{fannjiang2020autofocused, chan2021deep} suggest using a reverse mapping for predicting the input design from the property score using generative modeling techniques. By maintaining consistency between the forward mapping (i.e., the model prediction process) and the reverse mapping, the model's accuracy is significantly enhanced. A parallel strategy can be applied in explicit proxy updates where the introduction of a reverse mapping to predict the outer variable from the inner variable could improve the accuracy of the proxy by ensuring consistency.
> > > > > > > >
> > > > > > > > **Efficient Sampling.** Training the proxy neural network involves sampling the outer variable and computing the corresponding inner variable, which can be computationally expensive. Utilizing the acquisition function, as seen in model-based optimization \citep{trabucco2022design}, to decide the next batch for sampling may prove to be an efficient solution to this challenge.

---

### Decision · Action_Editors · 2023-07-15

**Recommendation:** Reject

**Comment:**

TMLR's guidelines suggest that "We want survey papers that draw new, previously unreported connections between several pieces of work in an area, and/or that clearly highlight trends in the area and/or suggest currently open problems."

The authors made efforts in revising their paper, in order to reflect reviewers' comments. The revised version seems to be better than the original submission. However, all of reviewers criticized that the paper is still unclear, structured in a confusing way, and provides very little insight despite being almost 20 pages long. Thus, I have no other choice but to reject it.

**Audience:**

The topic itself might be of interest to many people in TMLR's audience. However, reviewers pointed out that the paper does not provide a clear introduction to the topic.

**Claims And Evidence:**

This is a review paper which provides a survey of many methods for bi-level optimization. It covers a lot of work using bi-level optimization, which is a strength of this paper. However, its weakness is its lack of clarity, insights, or new knowledge.